# Relatively warm deep-water formation persisted in the Last Glacial Maximum

Jack H. Wharton[1 ✉], Emilia Kozikowska[1,2], Lloyd D. Keigwin[3], Thomas M. Marchitto[4], Mark A. Maslin[1], Martin Ziegler[2] & David J. R. Thornalley[1,3]

The Last Glacial Maximum (19–23 thousand years ago) was characterized by low greenhouse gas concentrations and continental ice sheets that covered large parts of North America and Europe[1]. Glacial climate was therefore very different, with colder global mean temperatures and an increased Equator-to-pole temperature gradient, probably resulting in stronger westerlies[2]. However, the state of the deep North Atlantic Ocean under these glacial climate forcings remains uncertain[3–6], particularly owing to the rarity of deep-ocean temperature and salinity constraints. Here we show that the temperature of the glacial deep (>1.5 km) Northwest Atlantic was approximately 0–2 °C (only 1.8 ± 0.5 °C (2 s.e.) colder than today), and, after accounting for the whole-ocean change, seawater $\delta^{18}O$ was 0.3 ± 0.1‰ (2 s.e.) higher and can be traced back to the surface subtropics via the subpolar Northeast Atlantic and Nordic Seas. Together, our hydrographic data reveal the thermal and isotopic structure of the deep Northwest Atlantic and suggest sustained production of relatively warm and probably salty North Atlantic Deep Water during the Last Glacial Maximum. Furthermore, our results provide updated constraints for benchmarking Earth system models used to project future climate change.

At present, the deep North Atlantic Ocean is predominantly thermally stratified, with relatively warm (2–4 °C) and salty (34.9 practical salinity units (PSU)) North Atlantic Deep Water (NADW) occupying depths between approximately 1 km and 4 km, and colder (0–1 °C) and fresher (34.7 PSU) Antarctic Bottom Water (AABW) below[7,8] (Fig. 1). The hydrographic properties (temperature and salinity) of both NADW and AABW are determined by the processes that govern their formation. NADW is formed from the cooling and densification of relatively warm and salty surface Atlantic waters sourced from the western subtropics and carried northeastwards via the North Atlantic Current (NAC). By contrast, AABW is formed from fresher waters on the continental shelves around Antarctica via intense cooling and brine rejection from sea-ice growth[9].

It remains uncertain whether and how the hydrographic structure of the deep North Atlantic changed during the Last Glacial Maximum (LGM). Modelling results suggest that the glacial climate and ice sheets would have driven stronger wind-driven gyre circulation, resulting in increased northwards salt transport and oceanic heat loss over the subpolar gyre, both of which would have favoured enhanced NADW production[10,11]. Conversely, increased glacial sea ice and calving glaciers would have reduced oceanic heat loss, potentially supressing deep-water formation[12]. Traditionally, palaeoceanographic nutrient proxies have been used to infer a glacial shoaling of the boundary between glacial NADW and AABW to around 2 km (ref. 3). However, recent modelling studies using stable carbon and oxygen-isotope ratios ($\delta^{18}O$ ($^{18}O/^{16}O$) and $\delta^{13}C$ ($^{13}C/^{12}C$)), nutrient and carbonate-ion concentration proxies (that is, Cd/Ca and B/Ca), and water-mass indicators ($\epsilon$Nd ($^{143}Nd/^{144}Nd$);

albeit with uncertainties regarding non-conservative behaviour and end-member non-stationarity) suggest this may be an overestimate[4,5,13]. Recent results have also revealed a deeper northern subtropical gyre and associated subtropical mode waters (STMWs)[14], and an abyssal deep-water mass of northern origin below approximately 5 km (ref. 15).

The associated hydrographic properties of the deep North Atlantic during the LGM are also uncertain—at present, palaeoceanographic reconstructions of temperature and salinity are limited to a small number of discrete sites. Among these, the most frequently cited are derived from sedimentary pore waters, which suggest that the glacial equivalents of NADW and AABW were both near freezing (−1.1 °C and −1.9 °C, respectively), with variations in salinity—rather than temperature—driving glacial stratification[6]. However, further investigations using inverse modelling have queried the assumptions underpinning these data (for example, the use of the prior condition that temporal changes in regional deep-ocean salinity scale with global benthic foraminiferal $\delta^{18}O$ and mean sea-level history), highlighting the need to attribute larger uncertainties to these pore-water-based temperature and salinity estimates[16,17]. Therefore, new palaeoceanographic proxy reconstructions are necessary to better constrain the hydrographic properties of the deep North Atlantic during the LGM.

Here we present geochemical reconstructions of seawater temperature and $\delta^{18}O$ (henceforth, $\delta^{18}O_{sw}$) from 13 marine sediment cores collected at Cape Hatteras, Blake Outer Ridge, Bermuda Rise and Corner Rise in the Northwest Atlantic. These cores form a depth transect spanning water depths between approximately 1.5 km and 5 km, and are complemented by 3 additional cores retrieved from south of Iceland

[1]Department of Geography, University College London, London, UK. [2]Department of Earth Sciences, Utrecht University, Utrecht, Netherlands. [3]Woods Hole Oceanographic Institution, Woods Hole, MA, USA. [4]Department of Geological Sciences and INSTAAR, University of Colorado, Boulder, CO, USA. ✉e-mail: jack.wharton.15@ucl.ac.uk

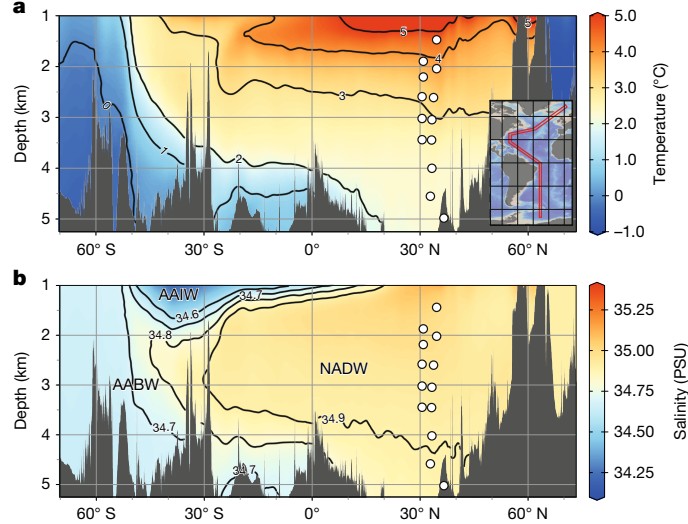

**Fig. 1 | Modern Atlantic temperature and salinity. a,b,** Meridional sections showing the thermal (**a**) and haline (**b**) structure of the Atlantic. The inset in **a** shows the transect (red box) used to derive these sections. Hydrographic data are from World Ocean Atlas 2023 (WOA23)[7,8] and were plotted using Ocean Data View[44]. Open circles show the locations of sediment cores from the Northwest Atlantic used in this study. Modern deep-ocean water-mass geometry is well resolved in salinity space (**b**). AAIW, Antarctic Intermediate Water.

in the Northeast Atlantic (Fig. 1, Extended Data Fig. 1 and Extended Data Table 1). Not only do these data provide insights into the hydrography of the deep North Atlantic during the LGM but also they offer a means to constrain its vertical structure, given that both temperature and $\delta^{18}O_{sw}$ behave as conservative tracers.

At present, our Northwest Atlantic sites are predominantly bathed by NADW, which is composed of Labrador Sea Water (LSW)—formed in the Irminger, Iceland and Labrador seas—and the downstream products of Iceland–Scotland Overflow Water and Denmark Strait Overflow Water, both formed in the Nordic Seas[18]. Our Northwest Atlantic core sites thus encompass the main export pathway for deep waters formed across the subpolar North Atlantic. In addition, sites from <2.5 km at Blake Outer Ridge allow us to reconstruct the properties of the deep subtropical gyre and associated STMWs, which extended to depths of 2 km to 2.5 km during the LGM[14]. Our Northeast Atlantic cores are situated along the main flow path of Iceland–Scotland Overflow Water

as it transits through the Iceland Basin before combining with LSW and Denmark Strait Overflow Water to form NADW.

We reconstructed deep-ocean temperatures from the North Atlantic during the mid-to-late Holocene (2–6 thousand years ago (ka) before present (BP)) and the LGM by measuring trace-metal ratios (Mg/Ca and Mg/Li) in multiple species of benthic foraminifera (Methods), both of which are positively correlated with seawater temperature during calcification. In particular, we focused on aragonitic species, and for calcitic taxa, we focused on infaunal species, whose magnesium (Mg) partitioning during calcification is thought to be less affected by a low carbonate-ion saturation state ($\Delta CO_3^{2-}$; Methods), as they calcify under the influence of surrounding pore waters, which can be buffered[19]. For the aragonitic species, *Hoeglundina elegans*, and the deep infaunal species, *Globobulimina affinis*, we converted Mg/Li and Mg/Ca to temperature using published calibrations[20,21], respectively (Fig. 2). For *Melonis* spp. and *Cassidulina neoteretis*, we developed a single core-top calibration using data from previous calibration studies (Extended Data Fig. 2), which show that numerous low-Mg, shallow infaunal benthic foraminifera show similar temperature sensitivities (approximately 0.1 mmol mol$^{-1}$ °C$^{-1}$; Fig. 2 and Methods). The resultant temperature estimates from these different species are consistent and directly comparable, indicating no significant inter-species bias. We therefore averaged these multi-species data to derive mid-to-late Holocene and LGM mean-temperature estimates for each core, reducing the overall uncertainty (Methods). As $\Delta CO_3^{2-}$ may not be completely buffered in sedimentary pore waters, we also generated additional independent temperature estimates by measuring benthic foraminiferal clumped isotopes ($\Delta_{47}$; Methods and Extended Data Fig. 3). Both trace-metal- and $\Delta_{47}$-based temperature estimates were then combined with paired stable oxygen-isotope measurements[14] to derive independent trace-metal- and $\Delta_{47}$-based estimates of $\delta^{18}O_{sw}$ (Methods), which is correlated with salinity in the modern ocean.

## Temperature and $\delta^{18}O_{sw}$ reconstructions

Mid-to-late Holocene multi-proxy temperature and $\delta^{18}O_{sw}$ estimates from similar depths between 1 km and 4.5 km show good agreement, generally showing temperatures and $\delta^{18}O_{sw}$ ranging from 2 °C to 4 °C and from 0‰ to 0.25‰, respectively (Fig. 3a,b). These data also agree well with both modern observational data[7,8] and ostracod-based Mg/Ca temperature estimates[22–24] (Fig. 3a) from the mid-to-late Holocene. If we calculate density using temperature and a single $\delta^{18}O_{sw}$–salinity relationship, this incorrectly results in a density inversion with depth, with some sites offset from modern observations[25] (Fig. 3c and Methods).

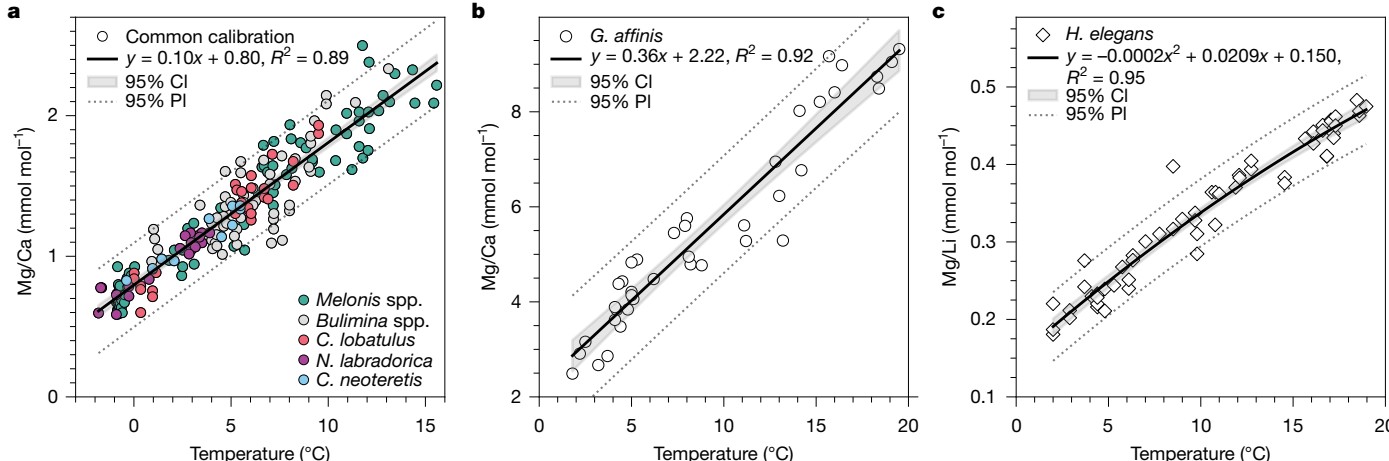

**Fig. 2 | Benthic foraminiferal trace-metal temperature calibrations used in this study. a–c,** Trace metal versus estimated growth temperature for multiple species of benthic foraminifera that all show similar temperature sensitivities

(**a**; Methods, Extended Data Fig. 2 and references therein), *G. affinis*[20] (**b**) and *H. elegans*[21] (**c**). The shading and dashed lines represent 95% confidence (CI) and 95% prediction (PI) intervals, respectively.

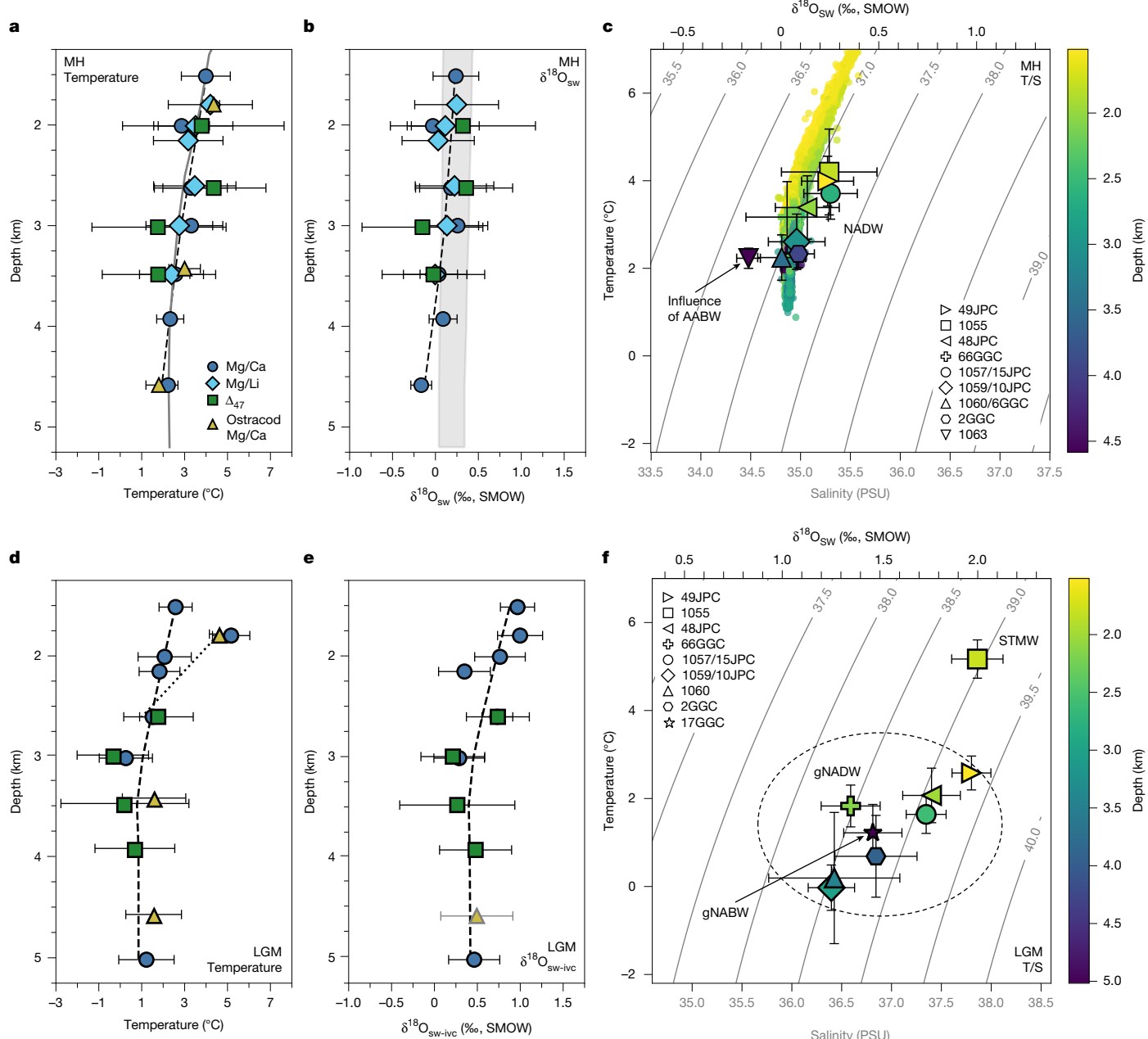

**Fig. 3 | Hydrographic structure of the Northwest Atlantic during the mid-to-late Holocene and LGM. a–f,** Vertical temperature (**a**,**d**) and $\delta^{18}O_{sw}$ (**b**,**e**) profiles, and temperature versus salinity (T/S) cross-plots (**c**,**f**) for the mid-to-late Holocene (MH; **a**–**c**) and the LGM (**d**–**f**). In **b** and **f**, the use of grey versus black axis labels denotes weaker (grey) versus more robust (black) proxy reconstruction. The filled coloured symbols in **a**, **b**, **d** and **e** represent the mean value for each depth (individual and mean monospecific temperature data and are shown in Extended Data Fig. 4), and associated errors bars are ±2 s.e. (Methods). All $\delta^{18}O_{sw}$ data are reported relative to the Standard Mean Ocean Water (SMOW) scale. The dashed black lines are locally weighted scatterplot smoothing lines (smoothing span, 1) through all foraminiferal temperature data from this study. The grey line and ribbon in **a** and **b**, respectively, denote the modern temperature from WOA23[7] and the $\delta^{18}O_{sw}$ structure of the Northwest Atlantic (in the absence of modern in situ $\delta^{18}O_{sw}$ measurements, a range of $\delta^{18}O_{sw}$ was derived using salinity data from WOA23 (ref. 8) and modern salinity–$\delta^{18}O_{sw}$ relationships (NADW, North Atlantic (NATL) and LSW)[25]). The dotted best fit

line in **d** shows the shift to warmer temperatures, most probably owing to the influence of a deeper glacial subtropical gyre at Blake Outer Ridge[14]. Ostracod temperature data are derived from published benthic ostracod shell Mg/Ca ratios[22–24]. The glacial $\delta^{18}O_{sw}$ estimate at approximately 4.5 km is derived using published ostracod Mg/Ca temperature data and nearby published benthic foraminiferal $\delta^{18}O$ data[45,46]. Symbol colours in **c** and **f** correspond to core water depth and associated errors are ±1 s.e. (Methods). Isopycnals of $\sigma_2$ were calculated using modern temperature and salinity measurements from the Global Ocean Data Analysis Project (GLODAP, v2.2022)[47] and the Gibbs seawater Oceanographic Toolbox (TEOS-10 standard)[48]. North Atlantic (20–60° N, 0–80° W) GLODAP (v2.2022) temperature and salinity measurements are also plotted as smaller coloured circles and coloured according to water depth in **c**. To aid comparison, **c** and **f** are offset by 1.1 PSU to account for the LGM–Holocene whole-ocean salinity difference, derived from the change in global sea level. gNADW, glacial North Atlantic Deep Water; gNABW, glacial North Atlantic Bottom Water.

This is because different $\delta^{18}O_{sw}$–salinity relationships are applicable to the various subcomponents of NADW (for example, LSW versus over-flow waters). To avoid this uncertain complexity, we do not make use of

a density conversion to the mid-to-late Holocene data, nor is it possible to accurately do so for the glacial data either. Overall, these data imply that a modern-like circulation was prevalent during the mid-to-late

Holocene, with relatively warm and salty NADW present down to at least 4 km in the Northwest Atlantic. Given that our multi-proxy data appear to be faithfully capturing in situ deep-ocean temperature and $\delta^{18}O_{sw}$, we now apply them to the LGM.

Apart from site ODP-172-1055 (approximately 1.8 km), which is substantially warmer (about 5 °C), glacial temperature and $\delta^{18}O_{sw}$ reconstructions are relatively uniform, ranging from 0 °C to 2 °C and from 1.25‰ to 1.75‰, respectively (Fig. 3d,e). Similar to the mid-to-late Holocene, there is strong agreement—within the margin of error—between our Mg/Ca- and $\Delta_{47}$-based temperature and $\delta^{18}O_{sw}$ estimates, as well as with the few other independent data from the Northwest Atlantic[22–24]. In particular, the relative warmth of site ODP-172-1055 (33° N) is consistent with the influence of a deeper subtropical gyre extending down to below approximately 2 km during the LGM[14] (Fig. 3d,f). Our deeper data are also consistent with previous work that suggests that the glacial deep Northwest Atlantic was dominated by NADW[5,13], rather than being occupied by distinct northern- and southern-source waters. If we calculate glacial densities using the modern NADW $\delta^{18}O_{sw}$–salinity relationship, this again results in a density inversion (Fig. 3f), which implies that the glacial North Atlantic was filled with multiple modes of glacial NADW[5], probably formed at different locations and characterized by different $\delta^{18}O_{sw}$–salinity relationships. For example, our data from sites at 3–4 km hint at being colder and having lower $\delta^{18}O_{sw}$, which may be consistent with a deep-water-formation region more affected by sea-ice formation and brine-rejection processes, such as proposed for the glacial Arctic Mediterranean[5]. Furthermore, our abyssal (>5 km) constraints indicate a temperature and $\delta^{18}O_{sw}$ similar to our other deep North Atlantic data, which is consistent with the inference from previous carbon-isotope ($\delta^{13}C$ and $^{14}C$) evidence of a northern-origin abyssal water mass in the Northwest Atlantic during the LGM[15].

## Warm and salty glacial North Atlantic

Notably, our reconstructed glacial temperatures from depths currently bathed by NADW (1.5–4 km, excluding ODP-172-1055) in the Northwest Atlantic, are on average less than 2 °C colder than modern temperatures at equivalent depths ($\Delta T_{Mg/Ca} = -1.63 \pm 0.44$ °C; $\Delta T_{(\Delta47)} = -2.02 \pm 0.95$ °C; Fig. 4a). In addition, glacial temperature constraints from south of Iceland also suggest similarly modest cooling in the Northeast Atlantic ($\Delta T_{Mg/Ca} = -1.78 \pm 0.56$ °C; Fig. 4a). These data stand in contrast to glacial estimates derived from sedimentary pore waters[6], which suggest near-freezing conditions at 2 sites (2.2 km and 4.6 km; Fig. 4a); thus, our results instead indicate that the deep North Atlantic remained relatively warm (approximately 0–2 °C) during the LGM.

In comparison, average-ice-volume-corrected glacial $\delta^{18}O_{sw}$, hereafter $\delta^{18}O_{sw-ivc}$ (Methods), is higher than equivalent modern $\delta^{18}O_{sw}$ in the Northwest Atlantic ($\Delta\delta^{18}O_{sw-ivc(Mg/Ca)} = 0.38 \pm 0.14$‰; $\Delta\delta^{18}O_{sw-ivc(\Delta47)} = 0.23 \pm 0.24$‰; Fig. 4b) and Northeast Atlantic ($\Delta\delta^{18}O_{sw-ivc(Mg/Ca)} = 0.51 \pm 0.19$‰). This suggests that previous glacial deep North Atlantic $\delta^{18}O_{sw}$ estimates derived from sedimentary pore waters were too low, probably owing to the greater methodological uncertainties and assumptions now recognized with this method, when applied to the North Atlantic[16,17]. Therefore, although deriving reliable palaeosalinity estimates for the North Atlantic during the LGM is challenging—owing to the spatial and temporal variability in the $\delta^{18}O_{sw}$–salinity relationship[26]—higher glacial $\delta^{18}O_{sw-ivc}$ is probably due to a combination of variable $\delta^{18}O_{sw}$–salinity relationships and saltier NADW relative to the Holocene.

We also compared our $\delta^{18}O_{sw-ivc}$ reconstructions with the limited number of available isotope-enabled LGM simulations, which produce a relatively wide range of $\delta^{18}O_{sw-ivc}$ values for NADW (approximately −0.2‰ to 0.5‰; Methods and references therein). Of these, the iPOP2 (Parallel Ocean Program version 2) simulation[27] shows the best agreement with our proxy data, simulating high near-surface

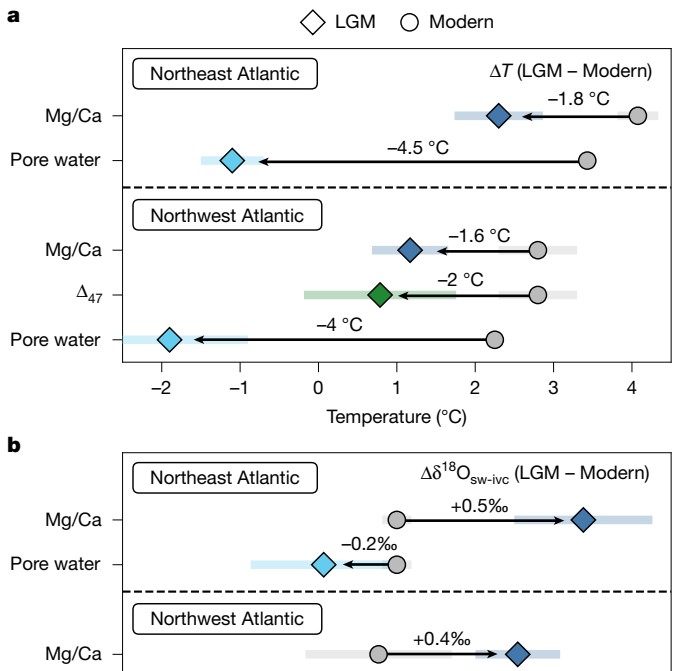

**Fig. 4 | Summary of deep-ocean temperature and $\delta^{18}O_{sw}$ changes in the North Atlantic between the LGM and the modern ocean. a,b,** Calculated difference in deep-ocean temperatures ($\Delta T$; **a**) and $\delta^{18}O_{sw-ivc}$ ($\Delta\delta^{18}O_{sw-ivc}$; **b**) derived from benthic foraminiferal Mg/Ca- and $\Delta_{47}$-based estimates (dark blue and green, respectively) and published sedimentary pore-water $\delta^{18}O$-based estimates (light blue[6]; Methods; potential temperature was converted to in situ temperature using the Gibbs seawater Oceanographic Toolbox (TEOS-10 standard[48]); $\delta^{18}O_{sw}$ is reported relative to the SMOW scale). For the Northwest Atlantic, we exclude data from the subtropically influenced site ODP-172-1055 and the abyssal site KNR-197-10-17GGC. Glacial Northeast Atlantic data are from sediment cores RAPiD-10-1P, RAPiD-17-5P and BOFS17K (Extended Data Table 1). Associated errors bars are ±2 s.e. (Methods). Modern temperature data were taken from WOA23 (ref. 7) (grey), and in the absence of equivalent $\delta^{18}O_{sw}$, we assume a modern $\delta^{18}O_{sw}$ of 0.20 ± 0.2‰ and 0.25 ± 0.05‰ for the Northwest and Northeast Atlantic, respectively (for example, ref. 47). As in Fig. 3, glacial $\delta^{18}O_{sw}$ has been corrected by −1.0‰ to account for changes in global ice volume (Methods). We also calculated $\Delta T$ and $\Delta\delta^{18}O_{sw-ivc}$ for the LGM and mid-to-late Holocene using paired samples where available; the resulting estimates show close agreement with our broader climatological comparison ($\Delta T_{Mg/Ca} = -1.7 \pm 0.7$ °C, $\Delta\delta^{18}O_{sw-ivc(Mg/Ca)} = 0.5 \pm 0.2$‰, $n = 5$; $\Delta T_{(\Delta47)} = -2.2 \pm 2.0$ °C, $\Delta\delta^{18}O_{sw-ivc(\Delta47)} = 0.4 \pm 0.5$‰, $n = 3$; Methods and Source Data).

$\delta^{18}O_{sw-ivc}$ values in the western subtropical North Atlantic (1‰), which feed through into NADW at depth. However, neither STMW nor NADW extend as deep in the simulations compared with proxy reconstructions[5,14], probably owing to limitations in the ability of models to simulate deep-water-formation processes, in part linked to their spatial resolution[28].

## Sustained glacial deep-water production

A comparison of our glacial deep-ocean $\delta^{18}O_{sw-ivc}$ data with equivalent glacial data from sites along the modern pathway of NADW and its near-surface source waters reveals that the high $\delta^{18}O_{sw-ivc}$ signature of glacial NADW can be traced along that same route. Figure 5 shows how high $\delta^{18}O_{sw-ivc}$—recorded consistently by multiple planktic foraminiferal

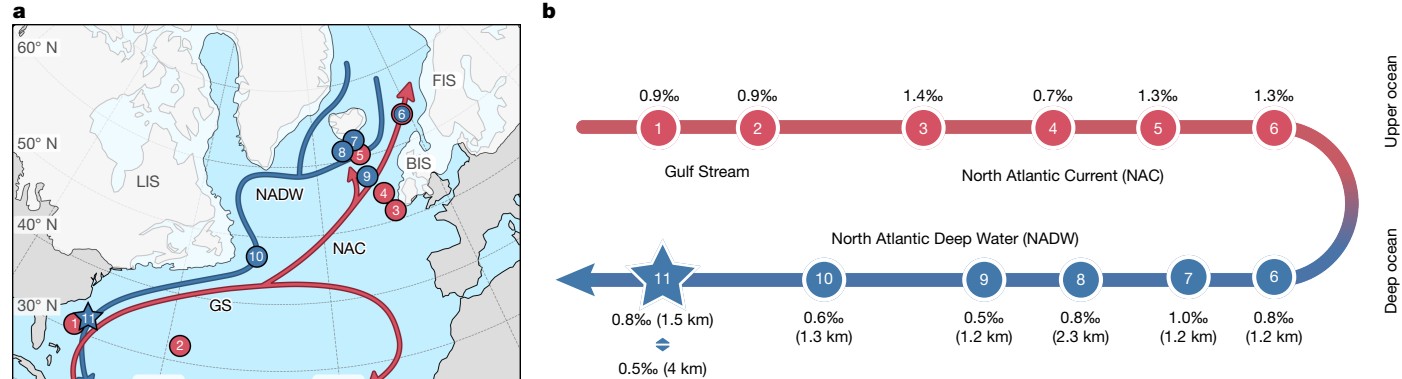

**Fig. 5 | High δ¹⁸O_sw-ivc signature reconstructed at multiple sites in the North Atlantic reveals the upstream pathway of glacial Northwest Atlantic deep waters. a**, Map showing the location of core sites used to reconstruct surface-ocean (red filled circles) and deep-ocean (blue filled circles) δ¹⁸O_sw-ivc from the glacial North Atlantic (Methods, Extended Data Table 3 and references therein, and Source Data). The numbered star (11) denotes the approximate position of our Northwest Atlantic transect). The red and blue arrows denote surface- and deep-ocean currents, respectively. White shaded areas denote the approximate extent of the Laurentide Ice Sheet (LIS)[49] and Feno-Scandinavian Sheet (FIS) and British-Irish Sheet (BIS)[50] at 21.5 ka BP. GS, Gulf Stream. **b**, Simplified schematic showing the potential upstream pathway of high δ¹⁸O_sw-ivc NADW (δ¹⁸O_sw is reported relative to the SMOW scale). The numbers along the arrow correspond to the numbered core sites in **a**. For consistency and where necessary, both planktic and benthic Mg/Ca temperature data were recalibrated using new and/or updated calibrations and foraminiferal calcite (δ¹⁸O_c) was corrected using species-specific corrections[14] (Methods and Source Data).

species that occupy and reflect the properties of the subsurface upper ocean that is the source for NADW—is traceable from the western subtropical Atlantic, northeastwards along the path of the Gulf Stream and NAC into the likely deep-water-formation regions of the glacial subpolar North Atlantic and Nordic Seas, then back south at depth into the deep Northwest Atlantic. We therefore infer that there was sustained deep-water formation in the subpolar North Atlantic (and southern intermediate-depth Nordic Seas) during the LGM, which is consistent with most glacial climate model simulations (for example, ref. 10). Furthermore, given that glacial Antarctic Intermediate Water (Extended Data Fig. 5a) and equatorial Atlantic surface waters[29], both of which feed the subtropical North Atlantic, were characterized by lower δ¹⁸O_sw-ivc (−0.4‰ to 0.3‰ and 0.4‰ to 0.5‰, respectively), we infer that processes occurring in the subtropics contributed to the particularly high δ¹⁸O_sw-ivc signature along the Gulf Stream–NAC–NADW pathway.

## Subtropical hydroclimate forcing

As our reconstructed glacial δ¹⁸O_sw-ivc from the Northwest Atlantic is approximately 0.5‰ higher than the Holocene, its source waters must have been subject to additional enrichment by hydrological fractionation processes such as negative precipitation minus evaporation (P − E). In the western subtropical Atlantic, the balance between precipitation and evaporation is probably the predominant control on δ¹⁸O_sw (ref. 30), with negative P − E causing higher δ¹⁸O_sw. Therefore, we infer that the high glacial δ¹⁸O_sw-ivc reconstructed from the Gulf Stream region of the western subtropical gyre (Fig. 5) indicates that the regional P − E during the LGM was negative relative to the Holocene, owing to decreased precipitation and/or increased evapotranspiration. Such a glacial hydroclimatic regime is supported by terrestrial proxy data, indicating reduced precipitation over North America and the Caribbean[31,32], and climate models also consistently simulate negative P − E over the North Atlantic Subtropical Gyre during the LGM, primarily owing to increased evaporation driven by stronger, cold and dry glacial winds[2,14]. Although there are complexities in relating δ¹⁸O_sw to salinity[26], these model results suggesting lower P − E over the subtropical gyre provide support for our inference that the high δ¹⁸O_sw surface- and deep-water values are recording the higher glacial salinity of the NAC and NADW, and that glacial deep-water production was sustained by the continued supply of salty (and warm) upper-ocean waters to the subpolar North Atlantic[10,11].

## Surface–deep-ocean decoupling

In light of these relatively warm temperature constraints, a logical question that arises is why the deep North Atlantic did not cool further, given that much of the surface subpolar region was close to freezing during winter[29]. Today, the deep Nordic Seas are near the freezing point (−1 °C to −1.5 °C) owing to deep open-ocean convection, and thus have little potential for further cooling. Actually, reconstructed glacial bottom-water temperatures from the intermediate and deep Nordic and Arctic seas instead suggest temperatures of 0–2 °C (up to 3 °C warmer than today[33]; Extended Data Fig. 5c and references therein)—a consequence of expanded glacial sea ice limiting heat loss to the atmosphere and thereby reducing open-ocean convection. Previous work has shown that, despite the inferred reduction in Nordic Seas convection, the overflow of dense waters from the Nordic Seas into the subpolar North Atlantic persisted during the LGM[5], probably driven by processes such as brine rejection or supercooling under glacial ice shelves[34]. Although the overflow waters themselves were not colder than today, the slightly lower temperatures of their downstream products—as reconstructed in this study—probably reflect the entrainment of colder upper and intermediate waters within the glacial subpolar North Atlantic[35].

At present, most deep-water formation occurs south of the Greenland–Scotland Ridge, in the subpolar North Atlantic, through convection processes and entrainment[18], and palaeoceanographic evidence indicates that this subpolar overturning was sustained and/or strengthened during the LGM, with an overall southwards shift in the locus of deep-water formation[36]. This overturning would have been facilitated, in part, by the relatively high inferred salinity of the glacial NAC, which would have promoted deep convection before surface waters cooled to freezing point. In addition, simplified conceptual models have shown that as climate cools, it becomes increasingly difficult to form very cold NADW[37]. This is because the buoyancy flux associated with surface cooling is reduced at lower temperatures, and the upper water column becomes more haline stratified, which inhibits deep convection. Consequently, deep-water formation shifts farther south to relatively warmer regions, producing deep waters with temperatures around 2–3 °C—broadly consistent with our glacial deep-ocean temperature reconstructions. A more extreme version of this process is the cooling of water at the northern edge of the subtropical gyre, forming deeper STMW[38]. As STMW forms from warm subtropical waters, our

data showing STMW temperatures of approximately 5 °C imply substantial heat loss—highlighting the importance of this low-latitude deep-water formation in glacial meridional heat transport.

This concept of continued glacial NADW formation aligns with a recent multi-proxy synthesis that concludes that glacial NADW was composed of multiple deep-water masses sourced from different regions throughout the North Atlantic and Arctic oceans and formed by a variety of mechanisms[5]. Here we show that the integrated downstream product of these different formation modes is hydrographically homogeneous, characterized by temperatures of 0–2 °C and a $\delta^{18}O_{sw\text{-}ivc}$ of 0.25–0.75‰, the latter of which suggests that glacial NADW was probably also relatively salty compared with the modern.

## LGM and future ocean implications

Our temperature and $\delta^{18}O_{sw}$ transects reveal sustained production of relatively warm and probably salty NADW during the LGM. However, as $\delta^{18}O_{sw}$ serves only as an approximate indicator of salinity, it remains unclear whether the glacial ocean was thermally—as it is today—or haline stratified, as inferred from pore-water measurements[6]. In light of recent estimates that the mean ocean temperature (MOT) during the LGM was approximately 2.3 ± 0.5 °C (1σ) colder than today[39], our data—showing that NADW was only 1.8 ± 0.5 °C colder—imply that other deep-water masses such as AABW or Pacific Deep Water must have cooled by more than NADW to account for the overall MOT decrease. Furthermore, given that the densest class of modern AABW found close to Antarctica ($T_{AABW\text{-}ANT} = -0.5$ °C (ref. 7)) is already near the freezing point of seawater (approximately −2 °C), its capacity for further cooling is limited; therefore, Pacific Deep Water ($T_{PDW} = 1$–1.5 °C) seems a more likely candidate for greater cooling. However, not all AABW, such as that found in the South Atlantic, is as cold ($T_{AABW\text{-}SATL} = 0$–1 °C), and thus some portions probably had greater cooling potential. Alternatively, and/or in addition, an increased relative volume of AABW globally, such as in the deep Pacific, could explain the glacial reduction in MOT[40], despite the relatively modest cooling of NADW. Moreover, it has been suggested that NADW supercooling was a trigger for glacial inception (for example, ref. 41); however, the presence of relatively warm NADW in the glacial North Atlantic implies that other processes in the Southern Ocean may have had a more important role in thermally isolating Antarctica.

This study demonstrates that, despite the colder climate state of the LGM, there was sustained production of NADW that was only 1.8 ± 0.5 °C colder than today. This persistence was probably driven by continued buoyancy loss, enabled by sufficient cooling of a steady supply of warm, salty water transported to mid-to-high latitudes via the NAC, in part maintained by the wind-driven gyre circulation[10]. Oceanographic observations of the twenty-first century[42], and modelling studies of future climate scenarios[43], suggest that NADW production can occur at different locations than those that were typical during the twentieth century. The ability to accurately predict the future of NADW thus depends on climate models correctly simulating deep-water formation processes across a range of climatic and geographic settings—an area where our data can provide valuable test-bed constraints.

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

## Methods

### Age models and core sampling

The Northwest Atlantic Ocean samples are the same as those used by ref. 14; therefore, we adopt the same age models and same sampling strategies (Extended Data Table 1). For our Northeast Atlantic cores, we also use previously published age models[51,52]. Stratigraphic information for each core, including multi-species benthic and planktic foraminiferal $\delta^{18}O$ and age constraints, is also shown alongside our multi-proxy temperature estimates in Extended Data Fig. 6.

### Multi-species and multi-proxy approach

As no single species of benthic foraminifera is present across all depths between 1.5 km and 5 km, we used a multi-species approach, measuring trace-metal ratios in multiple different taxa. We also took a multi-proxy approach, reconstructing deep-ocean temperatures using independent techniques: $\Delta_{47}$ and trace-metal ratios, thus providing additional support for the individual proxy reconstructions.

### Trace-metal analyses

Monospecific benthic foraminiferal samples consisting of between 5 and 15 individuals (where possible) were picked from the >250-μm sediment size fraction (where foraminiferal abundances were low, we also picked from the >212-μm fraction). Foraminifera were then gently crushed between glass slides, after which approximately one-third of the material was removed for isotopic analyses (these data were previously reported[14]). The remaining material was then loaded into 500-μl Bio-Rad polypropylene microcentrifuge tubes that had been pre-leached in hot 10% hydrochloric acid and used for trace-metal analysis.

All samples underwent oxidative and reductive cleaning following established methods[53], before each individual sample was analysed for a suite of trace and minor elemental ratios on a Thermo Finnigan Element2 magnetic sector inductively coupled plasma mass spectrometer at the University of Colorado, Boulder, as described in ref. 54. Long-term $\pm1\sigma$ precision is 0.5% for Mg/Ca, 0.9% for Li/Ca and 4.2% for B/Ca. Mn/Ca, Al/Ca and Fe/Ca ratios were also measured to screen for potential contamination from detrital material and/or secondary phases. For low-Mg species (for example, *Melonis pompilioides*), samples with contaminant ratios >0.1 mmol mol$^{-1}$ were rejected. For high-Mg species, for example, *G. affinis*, this threshold was scaled accordingly[55]. However, in cases where contamination indicators were only marginally above threshold and/or Mn/Ca values were elevated, but Mg/Ca was in good agreement with multiple coeval samples with low contaminant ratios, these data were retained (individual measurements are provided in the Source Data).

We also omitted all trace-metal data from the shallow infaunal benthic foraminifera *Uvigerina peregrina*, despite suggestions that it is less affected by $\Delta CO_3^{2-}$ (ref. 56). This is because the global calibration poorly constrains the relationship between Mg/Ca and temperature ($R^2 = 0.68$)[57], particularly at the cold end of the calibration (<5 °C), which is most relevant to the deep ocean. The temperature sensitivity (approximately 0.07 mmol mol$^{-1}$ °C$^{-1}$) is also substantially lower than that of other shallow, low-Mg infaunal species and does not align well with our common calibration dataset (Extended Data Fig. 2f). As a result, *U. peregrina* Mg/Ca yielded implausibly warm glacial temperature estimates (>5 °C) at our core sites, which are not compatible with our clumped isotope temperature estimates from this species.

### Trace-metal temperature calibrations

Deep-ocean temperatures were reconstructed using benthic foraminiferal Mg/Ca and species-specific calibrations taking the form: Mg/Ca = $s \times T + c$, where $T$ is the calcification temperature in °C, and $s$ and $c$ are the calibration slope (temperature sensitivity) and intercept, respectively (Fig. 2a,b). One-sigma uncertainties on

individual monospecific temperature estimates were propagated, incorporating analytical error (assumed as a relative percentage error on Mg/Ca (see 'Trace-metal analyses')) and uncertainties in both the slope and intercept (Extended Data Fig. 4a,c,d).

Mid-to-late Holocene deep-ocean temperatures were also reconstructed using benthic foraminiferal Mg/Li, which has been shown to be less sensitive to carbonate-ion effects, especially in the aragonitic foraminifera, *H. elegans*[58]. Mg/Li-based temperature estimates were derived first by dividing Mg/Ca by Li/Ca and using the *H. elegans* temperature–Mg/Li calibration[21], which takes the form: Mg/Li = $aT^2 + bT + c$, where $T$ is the calcification temperature in °C and $a$, $b$ and $c$ are the quadratic, linear and intercept coefficients, respectively (Fig. 2c). One-sigma uncertainties on individual *H. elegans* temperature estimates were propagated using Monte Carlo analysis (10,000 iterations), in which each Mg/Li value was perturbed according to analytical uncertainty, and each of the 3 calibration coefficients was randomly sampled from a normal distribution defined by its respective $1\sigma$ uncertainty. The quadratic equation was solved for each perturbed Mg/Li value and coefficient set, yielding 10,000 plausible temperature estimates per sample. The $1\sigma$ standard deviation of these simulated temperatures was taken as the uncertainty for that sample (Extended Data Fig. 4b).

Mean mid-to-late Holocene and LGM trace-metal-based temperature estimates for each core were derived by first averaging data from the same species within each core. The uncertainty on this mean was expressed as a total $\pm2$ standard error (2 s.e.), combining the $\pm1$ standard error of the mean (s.e.m.), calculated as the variability among replicate samples ($n > 1$), with the average $1\sigma$ propagated temperature measurement uncertainty (Extended Data Fig. 4e–g). Where $n = 1$, the 2 s.e. approximates to the $2\sigma$ propagated temperature measurement uncertainty (we note that this $n = 1$ uncertainty estimate does not account for variability among replicate measurements and thus probably underestimates the true uncertainty). When multiple different species were present in a core, we calculated a multi-species mean by averaging the species-specific means. The associated uncertainty was determined by propagating the individual species-specific uncertainties to the multi-species mean, treating temperature estimates derived from different foraminiferal species as independent estimates. We also combined data from cores at similar water depths (for example, 1057 and 15JPC; 1059 and 10JPC; 1060 and 6GGC) by averaging their mean values and propagating uncertainties using the same approach (Fig. 3a,d).

### Common Mg/Ca temperature calibration

Previous calibration studies have shown that *Melonis* spp.[59], *Bulimina* spp.[60], *Cibicides lobatulus*[61], *Nonionella labradorica*[62] and *C. neoteretis*[63] show similar Mg/Ca temperature sensitivities (approximately 0.1 mmol mol$^{-1}$ °C$^{-1}$; Extended Data Fig. 2), suggesting a shared underlying mechanism governing Mg incorporation during calcification. Therefore, rather than selecting species-specific calibrations for different low-Mg, shallow infaunal species, we developed a common calibration for these five species using published core-top calibration data (Fig. 2a). To do this, we first corrected any non-reductively cleaned data[64], then converted each dataset to Mg/Ca and temperature anomalies ($\Delta$Mg/Ca and $\Delta T$) to place them on a common scale. We then performed simple linear regression on the combined dataset, revealing a temperature sensitivity of 0.1 mmol mol$^{-1}$ °C$^{-1}$ ($R^2 = 0.88$). Finally, we calculated the average intercept for each species, assuming a common temperature sensitivity of 0.1 mmol mol$^{-1}$ °C$^{-1}$. Species-specific intercepts are shown in Extended Data Table 2.

### Clumped isotope analyses ($\Delta_{47}$)

To provide independent support for our trace-metal-based temperature reconstructions, we generated additional temperature estimates by measuring benthic foraminiferal clumped isotopes ($\Delta_{47}$). These analyses were possible because of the high abundances of *G. affinis*, *U. peregrina*, *C. neoteretis*, *Cibicidoides pachyderma* and *H. elegans* in several of our

cores. As this technique also yields foraminiferal $\delta^{18}O$ ($\delta^{18}O_c$) data, we were also able to independently estimate paired $\delta^{18}O_{sw}$ with these same samples. The benthic foraminifera were clean and showed no sign of post-depositional alteration from authigenic carbonate precipitation and/or diagenetic overgrowths or contamination from nannofossils and organic material (scanning electron microscope images of representative benthic foraminifera are provided as Supplementary Information); therefore, they were not cleaned or crushed before analysis.

A total of 575 stable and clumped isotope analyses were performed across 30 runs at Utrecht University. Samples weighing approximately 95 µg and 135 µg were prepared for analysis using a Thermo Scientific Kiel IV carbonate preparation device coupled to a Thermo Scientific MAT 253 mass spectrometer (conventional dual inlet method), and a Thermo Scientific 253 Plus mass spectrometer using the long-integration dual-inlet method, respectively. Each of the 30 analytical runs consisted of 46 samples, including 24 carbonate standards—ETH-1, ETH-2 and ETH-3—in a 1:1:5 ratio. These ETH standards differ in their $\delta^{13}C$, $\delta^{18}O$ and $\Delta_{47}$ composition and were used to calibrate the sample measurements, correct for $\delta^{13}C$ and $\delta^{18}O$ drift, and calculate empirical transfer functions. $\Delta_{47}$ values were reported on the Inter-Carbon Dioxide Equilibrium Scale (I-CDES90)[65]. Additional carbonate standards, MERCK and IAEA-C2, were measured alongside the samples to monitor the long-term reproducibility of the Kiel–253 Plus system, which showed a post-correction reproducibility of 0.033‰. To account for potential temporal drift, each run included both mid-to-late Holocene and LGM samples, alternating between the two where possible. This approach minimized drift-related bias and focused on reconstructing $\Delta T$ between the two time periods, thereby reducing reliance on the accuracy of any single $\Delta_{47}$–temperature calibration.

In the Kiel IV, each sample was dissolved in phosphoric acid (104% $H_3PO_4$) at 70 °C, converting carbonate to carbon dioxide ($CO_2$). The evolved gas was cooled to −196 °C using two liquid-nitrogen traps to concentrate the $CO_2$ and remove excess water. It was then further purified at −40 °C using a PoraPakQ trap to eliminate possible organic contaminants. Negative pressure base lines were corrected for as described in refs. 66,67. Pressure base lines were recorded at various $m/z$ 44 intensities (0 V, 5 V, 10 V, 15 V, 20 V, 25 V) before each analytical run. The relationship between background signal and intensity was used to correct for nonlinearities in isotopologues measurements. Samples with extreme initial intensities (<11,000 V or >20,000 V) or high $\Delta_{47}$ standard deviations were excluded. Standardized $\Delta_{47}$ values were calculated using an empirical transfer function based on the offset between measured and accepted ETH values. To correct for subtle nonlinearity, an initial offset correction was applied using ETH-3 standards within a ±1,000 V range. For the MAT 253 Plus, most runs used ETH-3 data from two preceding and following runs to ensure at least ten standards in the target range.

$\delta^{18}O_c$ values were normalized to Vienna PeeDee Belemnite using 15 ETH-3 standards preceding and following each sample. Species-specific offsets from isotopic equilibrium, were corrected for using published species-specific offsets[14], and we also excluded any outliers based on corrected $\delta^{18}O_c$. We omitted data from samples with $\delta^{18}O_c < 4.00$‰ and $\delta^{18}O_c > 5.00$‰ and $\delta^{18}O_c < 2.15$‰ and $\delta^{18}O_c > 3.15$‰ for the LGM and mid-to-late Holocene, respectively. Raw $\Delta_{47}$ values for each core and time period were averaged and converted to temperature using the calibration equation of ref. 68: $\Delta_{47} = (0.0397 \pm 0.0011) \times 10^6/T^2 + 0.1518 \pm 0.0128$, where $T$ is temperature in °C. Associated errors are reported as ±2 s.e., with standard error calculated as $\sigma/\sqrt{n}$, where $\sigma$ is the standard deviation of $\Delta_{47}$ values and $n$ is the number of measurements for each core for each time period. For sites represented by two cores from similar water depths (for example, 1057/15JPC and 1059/10JPC), $\Delta_{47}$ values from both cores were averaged before temperature conversion. This calibration of ref. 68 was selected because (1) it yields mid-to-late Holocene temperature estimates that provide the best statistical match with modern in situ deep-ocean observations, whereas alternative calibrations

yield poorer statistical agreement and/or implausible temperature estimates (Extended Data Fig. 3b,c), and (2) it is based exclusively on foraminifera and was produced in clumped isotope laboratories that use analytical procedures identical to those applied in this study

## $\delta^{18}O_{sw}$

Mid-to-late Holocene and glacial $\delta^{18}O_{sw}$ were calculated for each core using multi-proxy mean-temperature data (Fig. 2a,d), paired with benthic foraminiferal $\delta^{18}O_c$ and the following linear equation: $(\delta_c - \delta_{sw} + 0.27) = -0.224 \pm 0.002 \times T + 3.53 \pm 0.02$, where $T$ is the calcification temperature in °C, $\delta_{sw}$ is $\delta^{18}O_{sw}$ on the Standard Mean Ocean Water scale, and $\delta_c$ is $\delta^{18}O_c$ on the PeeDee Belemnite scale[45]. Multi-species mean Mg/Ca- and Mg/Li-derived temperatures were combined with previously published paired $\delta^{18}O_c$ data[14], whereas $\delta^{18}O_{sw}$ derived from $\Delta_{47}$ temperatures used $\delta^{18}O_c$ measurements from clumped isotope analyses. $\delta^{18}O_c$ from benthic foraminiferal species known to calcify in disequilibrium with seawater (for example, *G. affinis*) was corrected using empirically derived, species-specific offsets[14]. For each core, the ±2 s.e. uncertainty on each $\delta^{18}O_{sw}$ estimate was calculated by propagating and combining uncertainties in quadrature, which included contributions from the ±2 s.e. uncertainty on the mean $\delta^{18}O_c$, the mean multi-proxy temperature estimate, and the slope and intercept of the $\delta^{18}O_{sw}$–temperature equation. When combining data from cores at similar water depths (for example, 1057 and 15JPC), we first averaged the species-specific $\delta^{18}O_c$ estimates and then calculated the ±2 s.e. using the same error propagation approach as used for our temperature estimates.

## $\delta^{18}O_{sw-ivc}$

To facilitate comparison with modern and mid-to-late Holocene $\delta^{18}O_{sw}$, we calculated ice-volume-corrected $\delta^{18}O_{sw}$ ($\delta^{18}O_{sw-ivc}$) for the LGM by subtracting 1.0‰ to account for the global-ice-volume effect. This correction is based on a global mean change in $\delta^{18}O_{sw}$ of $1.0 \pm 0.1$‰ (ref. 69), derived from sedimentary pore-water measurements, including non-Atlantic sites, which are considered more appropriate for estimating global mean glacial $\delta^{18}O_{sw}$ (ref. 16). Notably, this value also agrees well with other independent estimates ($0.94 \pm 0.18$‰ (ref. 70) and $1.05 \pm 0.2$‰ (ref. 71), based on simple numerical models of ice-sheet growth and benthic $\delta^{18}O$ change at polar sites, respectively). To further evaluate this correction, we also estimated the global mean $\delta^{18}O_{sw}$ change using the LR04 benthic foraminiferal stack[72] and the most recent estimate of glacial MOT[39]. Assuming a MOT change of $2.3 \pm 0.5$ °C and temperature sensitivity of 0.224‰ °C$^{-1}$ (ref. 45), a glacial-to-modern benthic foraminiferal $\delta^{18}O$ shift of 1.65‰, yields a global mean $\delta^{18}O_{sw}$ shift of $1.13 \pm 0.11$‰ (1$\sigma$). This estimate is consistent, within uncertainty, with published values, supporting our use of the canonical 1.0‰ correction to derive $\delta^{18}O_{sw-ivc}$ for the LGM.

## Temperature–salinity plots

Multi-proxy temperature and $\delta^{18}O_{sw}$ data were averaged to produce individual estimates for each core during the mid-to-late Holocene and LGM, with associated uncertainties expressed as ±1 s.e. after combining the error associated with each proxy (for example, Mg/Ca, Mg/Li, $\Delta_{47}$). For the mid-to-late Holocene, where $\Delta_{47}$ estimates were not available for all combined core pairs (for example, 1057/15JPC), multi-proxy data for both cores were averaged, and where multiple estimates from the same proxy were present (for example, Mg/Ca), their uncertainties were combined as described previously.

For the mid-to-late Holocene, $\delta^{18}O_{sw}$ and its associated uncertainty were then converted to salinity using a single empirically derived $\delta^{18}O_{sw}$–salinity relationship[25], taking the form $S = (\delta^{18}O_{sw} + c)/s$, where $S$ is salinity in PSU, and $s$ and $c$ are the calibration slope and intercept, respectively. For the LGM, $\delta^{18}O_{sw}$ was first corrected for the global-ice-volume effect, then converted to salinity using the same $\delta^{18}O_{sw}$–salinity relationship, before a global +1.1 PSU offset was applied to account for the higher mean salinity of the glacial ocean, enabling

direct comparison with modern salinity data. Whereas previous work has shown that different $\delta^{18}O_{sw}$–salinity relationships apply at different depths in the Northwest Atlantic[14], probably reflecting distinct deep-water masses sourced from different regions across the subpolar North Atlantic, for simplicity, we apply the NADW-specific relationship ($s = 0.51$, $c = 17.75$), as it is the most appropriate for the majority of our core sites, that is, using this relationship yields the best agreement between our reconstructions and modern observations (Fig. 3c). Therefore, we are cautious not to over-interpret the derived salinity and density values, and instead focus primarily on the underlying $\delta^{18}O_{sw}$ signal. Furthermore, owing to the large and regionally variable uncertainties in converting $\delta^{18}O_{sw}$ to salinity[73], we do not propagate uncertainty using the errors associated with the slope and intercept of the $\delta^{18}O_{sw}$–salinity relationship; instead, we convert $\delta^{18}O_{sw}$ uncertainties directly into salinity using a fixed slope and intercept.

Extended Data Figure 5 also includes published Atlantic glacial temperature and $\delta^{18}O_{sw}$ estimates[20,74–80]. However, given the uncertainty surrounding the glacial $\delta^{18}O_{sw}$–salinity relationships associated with the water masses bathing these sites, we do not convert these $\delta^{18}O_{sw}$ data to salinity. The salinity axis and isopycnals are shown to enable visualization for the Northwest Atlantic transect data originally shown in Fig. 2c,f.

### $\Delta T$ and $\Delta\delta^{18}O_{sw}$

Because we do not have a complete set of both mid-to-late Holocene and glacial temperature estimates for all our cores—in general, there are more mid-to-late Holocene data from shallower sites and glacial data from deeper sites—simply averaging all data from each time period would introduce an artificial cold bias. To avoid this, we calculated the mean LGM–modern deep-ocean temperature difference ($\Delta T$) based on the individual $\Delta T$ values for each core site. This was done by first determining the $\Delta T$ for each core from between 1.5 km and 4 km depth (excluding subtropically influenced site ODP-172-1055), using the closest World Ocean Atlas 2023 (WOA23) data point to each site (it is noted that each WOA23 data point represents the annual average based on all valid data from 1955 to 2022)[7]. These per-core $\Delta T$ values were then averaged to derive regional $\Delta T$ estimates from Mg/Ca-derived temperature data for the Northwest Atlantic and Northeast Atlantic. The associated uncertainty (±2 s.e.) was determined by combining the uncertainty associated with each core's $\Delta T$ in quadrature. For $\Delta_{47}$, $\Delta T$ was calculated as a weighted average, with the number of individual measurements per core used as weights. We note that the mean $\Delta T \pm 2$ s.e. (0.94 °C) is almost identical to the mean ±2 s.e. associated with the mean of all $\Delta_{47}$ glacial temperature estimates (0.96 °C, $n = 284$). We followed the same procedures to derive LGM–modern $\Delta\delta^{18}O_{sw}$; however, in the absence of modern in situ $\delta^{18}O_{sw}$ measurements, $\delta^{18}O_{sw}$ for each core was estimated using salinity data from WOA23 (ref. 8) and modern salinity–$\delta^{18}O_{sw}$ relationships for NADW, North Atlantic (NATL) and LSW[25]. These three estimates were then averaged to produce a modern $\delta^{18}O_{sw}$ value for each core. To check for a potential proxy–climatology bias, we also calculated $\Delta T$ and $\Delta\delta^{18}O_{sw}$ using paired LGM and mid-to-late Holocene data where available for both trace-metal-derived ($n = 5$) and $\Delta_{47}$-derived ($n = 3$) reconstructions. To do this, we followed the same procedure outlined above; however, for mid-to-late Holocene sites with both Mg/Ca- and Mg/Li-based temperatures, we used their combined mean values. For 6GGC, where only Holocene data are available, we compared these with LGM estimates from the nearby site 1060, which is also located at approximately 3.5 km water depth.

### Carbonate-ion effects

Previous work suggests that tthe bottom-water carbonate-ion ($\Delta CO_3^{2-}$) concentration affects the partitioning of Mg during calcification in epifaunal benthic foraminifera, with lower Mg/Ca where the carbonate-ion saturation state ($\Delta CO_3^{2-}$, defined as $\Delta CO_3^{2-} = CO_3^{2-}_{\text{in-situ}} - CO_3^{2-}_{\text{saturation}}$) is low or undersaturated ($<0$ μmol kg$^{-1}$)[56,81]. Although infaunal benthic

foraminiferal Mg/Ca are generally considered to be less susceptible to undersaturation, pore-water $\Delta CO_3^{2-}$ is spatially and temporally variable, such that infaunal species may also be affected by $\Delta CO_3^{2-}$ (refs. 20,82). To assess the potential impact of low $\Delta CO_3^{2-}$ on our infaunal benthic foraminiferal Mg/Ca ratios, we also reconstructed bottom-water $\Delta CO_3^{2-}$ using B/Ca ratios measured on *Cibicidoides wuellerstorfi* (where present) applying the calibration[81]: B/Ca = $1.14 \pm 0.048 \times \Delta CO_3^{2-} + 177.1 \pm 1.41$, where $\Delta CO_3^{2-}$ is the carbonate-ion saturation state of seawater in μmol kg$^{-1}$. One-sigma uncertainties were propagated, incorporating analytical error (see 'Trace-metal analyses') and uncertainties in both the slope and intercept (Extended Data Fig. 7). Modern Northwest Atlantic $\Delta CO_3^{2-}$ was calculated using temperature, salinity, total alkalinity and total dissolved inorganic carbon data from nearby Global Ocean Data Analysis Project (GLODAP, v2022) stations[47] with PyCO2sys[83].

Overall, reconstructed $\Delta CO_3^{2-}$ is generally $>0$ μmol kg$^{-1}$, suggesting that the glacial deep Northwest Atlantic was oversaturated with respect to $\Delta CO_3^{2}$ (Extended Data Fig. 7b). Although this implies that any $\Delta CO_3^{2-}$-effect-related suppression of Mg incorporation in our infaunal benthic foraminifera was probably minimal, $\Delta CO_3^{2-}$ at abyssal site KNR-197-10-17GGC is much lower ($-37 \pm 4$ μmol kg$^{-1}$, $n = 2$), and it is also possible that pore waters may be more undersaturated than the overlying bottom waters[20]. However, given that undersaturated conditions supress the Mg incorporation during calcification, any $\Delta CO_3^{2-}$ effect would bias Mg/Ca-derived temperatures towards colder values. Therefore, our reconstructed glacial deep-ocean temperatures may represent a conservative (cold) estimate and attempting to correct for any $\Delta CO_3^{2-}$ effect at this abyssal site would produce slightly warmer glacial temperatures.

### Isotope-enabled models

To provide additional context for our proxy reconstructions, we compared our $\delta^{18}O_{\text{sw-ivc}}$ estimates with available isotope-enabled simulations of the glacial Atlantic[27,84–88]. These models generally simulate a slightly shallower glacial Atlantic Meridional Overturning Circulation, with NADW temperatures ranging from approximately 1 °C to −2 °C (refs. 27,88) and $\delta^{18}O_{\text{sw-ivc}}$ between −0.2‰ and 0.5‰ (refs. 27,84–88). Although this is broadly consistent with the traditional view—based on palaeoceanographic nutrient proxies[3]—that the Atlantic Meridional Overturning Circulation shoaled during the LGM, and with pore-water-based estimates suggesting that glacial NADW was much colder[6], these simulations are not consistent with more recent work[5] and our temperature and $\delta^{18}O_{sw}$ constraints from the North Atlantic. However, the iPOP2 model does reproduce comparably high $\delta^{18}O_{sw}$ values, although these are restricted to the upper approximately 2 km of the North Atlantic[27].

### Published data

Glacial deep-ocean temperature and $\delta^{18}O_{sw}$ data and associated uncertainties were calculated following the same procedures used for our Northwest Atlantic data (Extended Data Table 3 and Source Data). Where appropriate, we used our common calibration to derive temperature, applying a 10% correction to any non-reductively cleaned trace-metal data (for example, Mg/Ca$_{\text{reduc.}}$ = (Mg/Ca)/1.1) and appropriate $\delta^{18}O_c$ offsets to benthic foraminiferal $\delta^{18}O_c$ from species that calcify in disequilibrium with the surrounding seawater[14]. Consistent with our previous approach, *U. peregrina*-based temperature and $\delta^{18}O_{sw}$ estimates were omitted from this compilation owing to concerns with the Mg/Ca temperature calibration of *U. peregrina*.

Glacial surface-ocean temperature and $\delta^{18}O_{sw}$ estimates and associated uncertainties were also calculated from published Mg/Ca and $\delta^{18}O_c$ data using species-specific Mg/Ca temperature calibrations (Extended Data Table 3), appropriate vital-effect corrections[89–91], and the updated $\delta^{18}O$–temperature relationship based on inorganic calcite precipitation experiments[91] (Source Data). Although recent Mg/Ca temperature calibrations for planktic foraminifera include non-thermal influences such as whole-ocean salinity and pH, they do not account for spatial

variations in local salinity[92]. For species with a known salinity effect on Mg/Ca (that is, *Globigerinoides ruber* and *Globigerina bulloides*), we iteratively solved for a salinity value that is self-consistent with the North Atlantic $\delta^{18}O_{sw}$–salinity relationship ($s = 0.55$, $c = 18.98$)[25]. For both *G. ruber* and *G. bulloides*, we assume a glacial pH of $8.2 \pm 0.2$ (refs. 92,93), recognizing that this conservative uncertainty is the primary contributor to the relativley large errors on the final temperature and $\delta^{18}O_{sw}$ estimates (Source Data).

## Data availability

The proxy data that support these findings are publicly available through the data repository Pangaea at https://doi.org/10.1594/PANGAEA.988210[94,95]. GLODAP Bottle Data (version2.2022)[47] and World Ocean Atlas data[7,8] were downloaded from https://glodap.info/index.php/merged-and-adjusted-data-product-v2-2022/ and https://www.ncei.noaa.gov/products/world-ocean-atlas, respectively. Figure 1 was generated using Ocean Data View software[44]. Source data are provided with this paper.

## Code availability

Python code used to perform statistical analyses as part of this study are freely available on Zenodo at https://doi.org/10.5281/zenodo.17733604 (ref. 96).

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

**Acknowledgements** We thank the WHOI Seafloor Samples Laboratory, including E. Roosen, M. Starr, A. Patterson and G. Nissen, and H. Kuhlmann at the IODP Bremen Core Repository for help with core sampling; B. Atkinson, D. Fairman, I. Patmore and J. Shilland for general laboratory assistance; A. van Dijk and D. Efting for technical support in the clumped isotope laboratory; L. van Maldegem for assistance in the trace-metal laboratory; M. Stanley for help with scanning electron microscopy imaging; M. Daëron, W. Gray, D. Hodell, J. Holmes, J. Rae, L. Roberts, V. Taylor and the UCL OACD group for discussions; M. Wheeler for helping to design Fig. 5; and C. D'Alton and M. Irving for additional design help. J.H.W. was supported by the London NERC Doctoral Training Partnership grant (grant number NE/L002485/1). This project has received funding from NERC Project ReconAMOC (NE/S009736/1), the Leverhulme Trust, NSF grants OCE-1304291 and OCE-2233080, and EU Project 101059547-EPOC. This paper reflects only the authors' views, and the European Union cannot be held responsible for any use that may be made of the information contained herein. For the purpose of open access, the author has applied a Creative Commons Attribution (CC BY) licence to any author accepted manuscript version arising.

**Author contributions** The project was conceived by J.H.W., M.A.M. and D.J.R.T. Marine sediment cores were collected by L.D.K. J.H.W. analysed and interpreted the trace-metal data, with contributions from T.M.M. and D.J.R.T. Clumped isotope data were generated by E.K. and analysed and interpreted by J.H.W., with contributions from E.K., M.Z. and D.J.R.T. Project supervision by D.J.R.T. J.H.W. wrote the first draft of the paper, with contributions from D.J.R.T. All authors contributed to discussion and the final version of the paper.

**Competing interests** The authors declare no competing interests.

**Additional information**
**Correspondence and requests for materials** should be addressed to Jack H. Wharton.

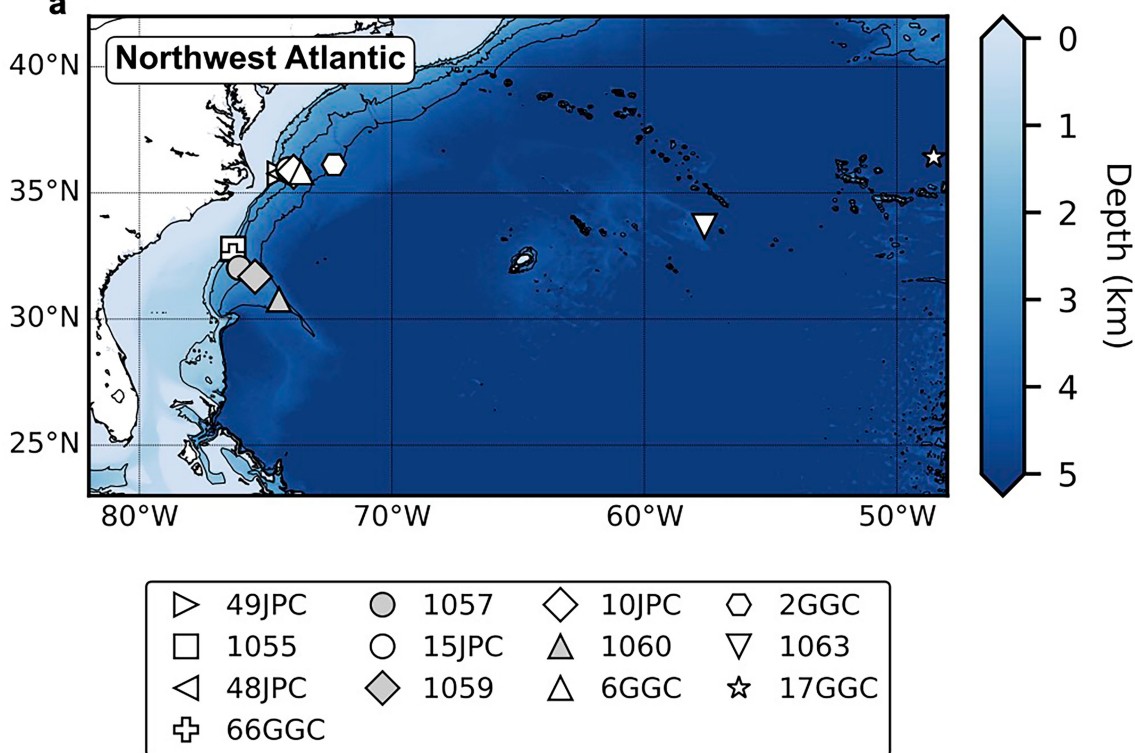

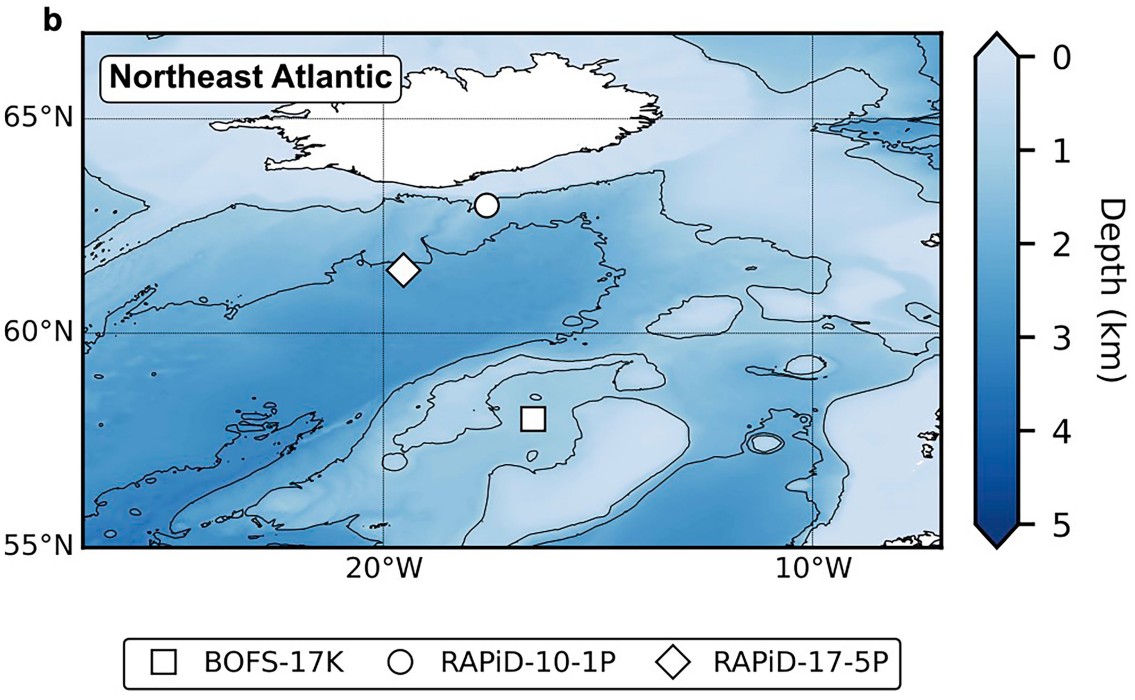

**Extended Data Fig. 1 | Map showing the location of cores used in this study.** Northwest Atlantic (**a**) and Northeast Atlantic (**b**). Ocean bathymetry is from the GEBCO_2014 global bathymetric grid (30 arc-second resolution)[97].

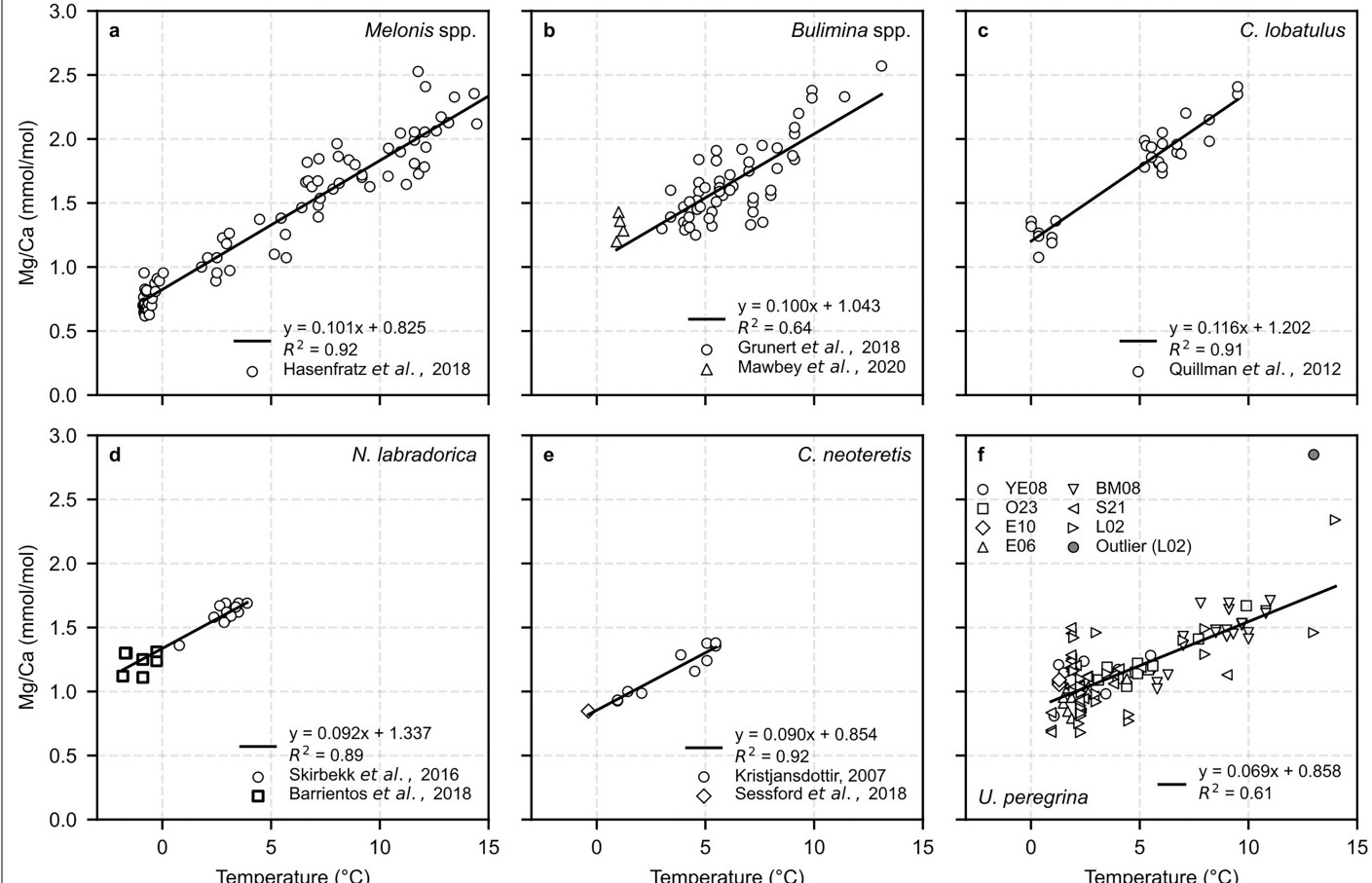

**Extended Data Fig. 2 | Species-specific benthic foraminiferal Mg/Ca-temperature calibration datasets used to develop our common calibration.** Published core-top Mg/Ca (mmol/mol) vs bottom water temperature (°C) are shown for six low-Mg, shallow infaunal benthic foraminifera: (**a**) *Melonis* spp.[59], (**b**) *Bulimina* spp.[60,98], (**c**) *C. lobatulus*[61], (**d**) *N. labradorica*[62,99], (**e**) *C. neoteretis*[63,100], and (**f**) *U. peregrina*[56–58,76,101,102]. Linear regressions were performed for each species using core-top data, often compiled from multiple studies. The resulting Mg/Ca-temperature relationships and associated $R^2$ values are shown. All species exhibit a temperature sensitivity of approximately 0.1 mmol/mol/°C, except *U. peregrina*, which shows a lower sensitivity (0.7 mmol/mol/°C). Inclusion of an outlier slightly increases the sensitivity

(~0.01 mmol/mol/°C,) however, it remains lower than the other five species associated temperature sensitives. For *C. neoteretis*, we excluded calibration data from ref. 99 due to the absence of a significant trend ($R^2 = <0.6$). Furthermore, including these data results in a lower temperature sensitivity (approximately 0.06 mmol/mol/°C), which yields implausibly cold temperature estimates for relatively high Mg/Ca ratios. In comparison, using data only from refs. 63,100, produces a higher temperature sensitivity more consistent with other shallow infaunal benthic foraminifera. Given the size of the *Melonis* spp. calibration dataset, we focused on data from the subtropical and subpolar North Atlantic (n = 82), most relevant to our study location.

**a**

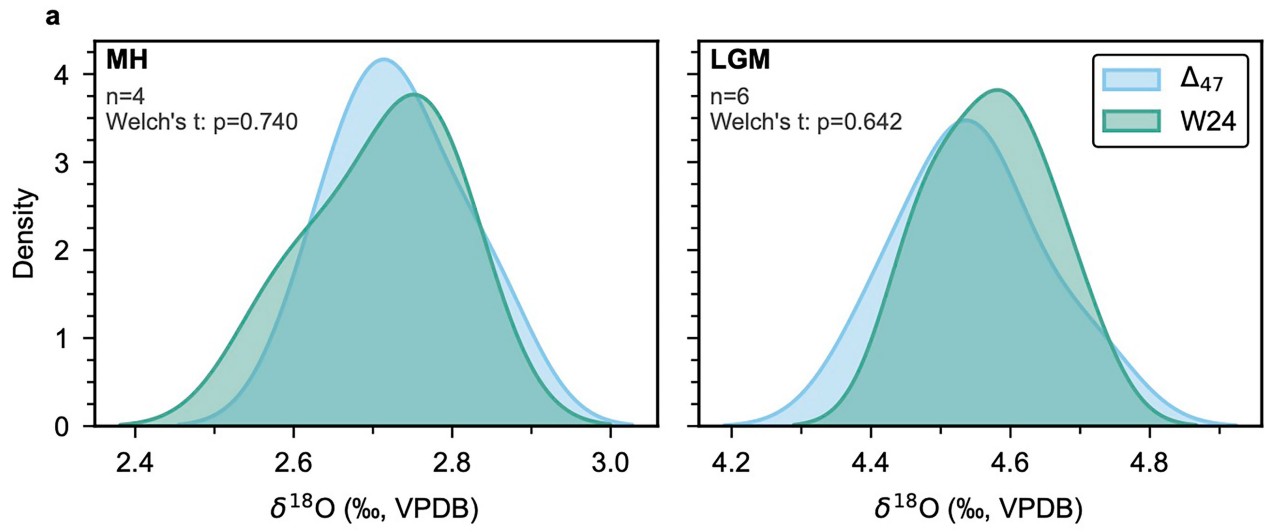

**b**

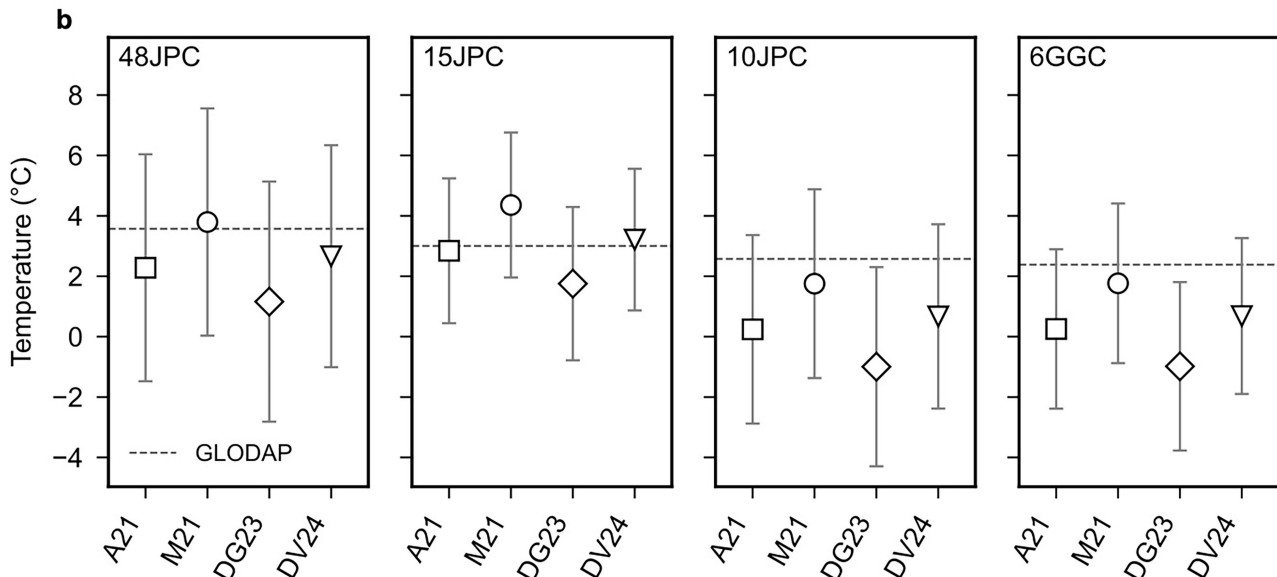

**c**

| Calibration | Per-core difference from modern (Reconstructed − modern; °C) | | | | Mean error (°C) | Mean absolute error (°C) |
|---|---|---|---|---|---|---|
| | 48JPC | 15JPC | 10JPC | 6GGC | | |
| A21 | -1.29 | -0.16 | -2.33 | -2.13 | -1.48 | 1.48 |
| M21 | 0.23 | 1.35 | -0.82 | -0.62 | **0.04** | **0.75** |
| DG23 | -2.41 | -1.25 | -3.57 | -3.37 | -2.65 | 2.65 |
| DV24 | -0.91 | 0.21 | -1.90 | -1.7 | -1.08 | 1.18 |

**Extended Data Fig. 3** | See next page for caption.

**Extended Data Fig. 3 | Comparison of δ¹⁸O generated as part of this study with equivalent published data and $\Delta_{47}$-based temperature estimates derived using different $\Delta_{47}$-temperature calibrations. a**. Kernel density estimate plots showing the distribution of $\delta^{18}O$ obtained from two independent analyses of benthic foraminifera from the same sediment samples for the mid-to-late Holocene and LGM ($\Delta_{47}$, this study, blue; stable isotope analysis as per ref. 14, teal). p-values from Welch's t-tests indicate no significant difference between the two datasets (in both cases p > 0.05), confirming that the analyses yield indistinguishable $\delta^{18}O$. Close agreement between the published data and published $\delta^{18}O$ data from around the Northwest Atlantic suggests that the $\delta^{18}O$ measured in this study were unaffected by contamination, e.g., from clays. **b** and **c**. Comparison of mid-to-late temperature estimates derived using different published $\Delta_{47}$-temperature calibrations: M21 (equation (1))[68], DG23 (equation (5))[103], A21 (equation (1))[104], and DV24 (equation (28))[105]. Dashed lines in **b** indicate the approximate modern in-situ temperature at each core site, based on GLODAP 2022 data[47]. The M21 calibration was selected for its strong statistical agreement with modern deep ocean temperatures and methodological consistency, despite being derived, in part, from planktic foraminiferal $\Delta_{47}$ data. This calibration was therefore applied to glacial samples as well.

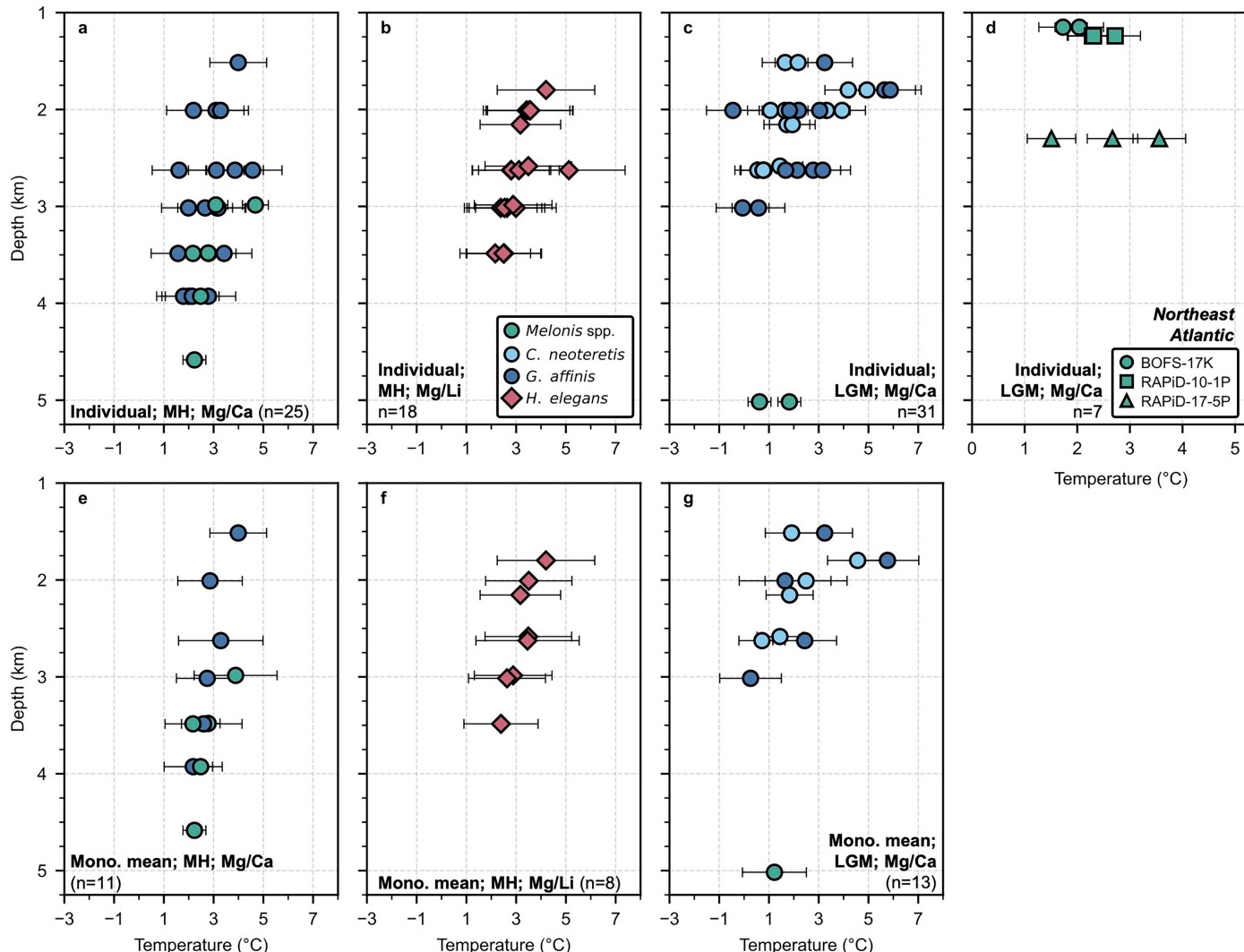

**Extended Data Fig. 4 | Individual and mean monospecific temperature data from the North Atlantic.** Individual Northwest Atlantic Mg/Ca and Mg/Li data for the mid-to-late Holocene (**a** and **b**) and LGM (**c**), and associated ±2 s.e. (2 s.e. = 2 × 1σ (T); methods). Each coloured circle denotes a single monospecific measurement, based on a sample comprising material from 5-15 specimens. **d**. same as for **c** but Northeast Atlantic data. **e-g**, same as **a-c** but each coloured circle denotes the monospecific mean for each core, with associated ±2 s.e. (Methods).

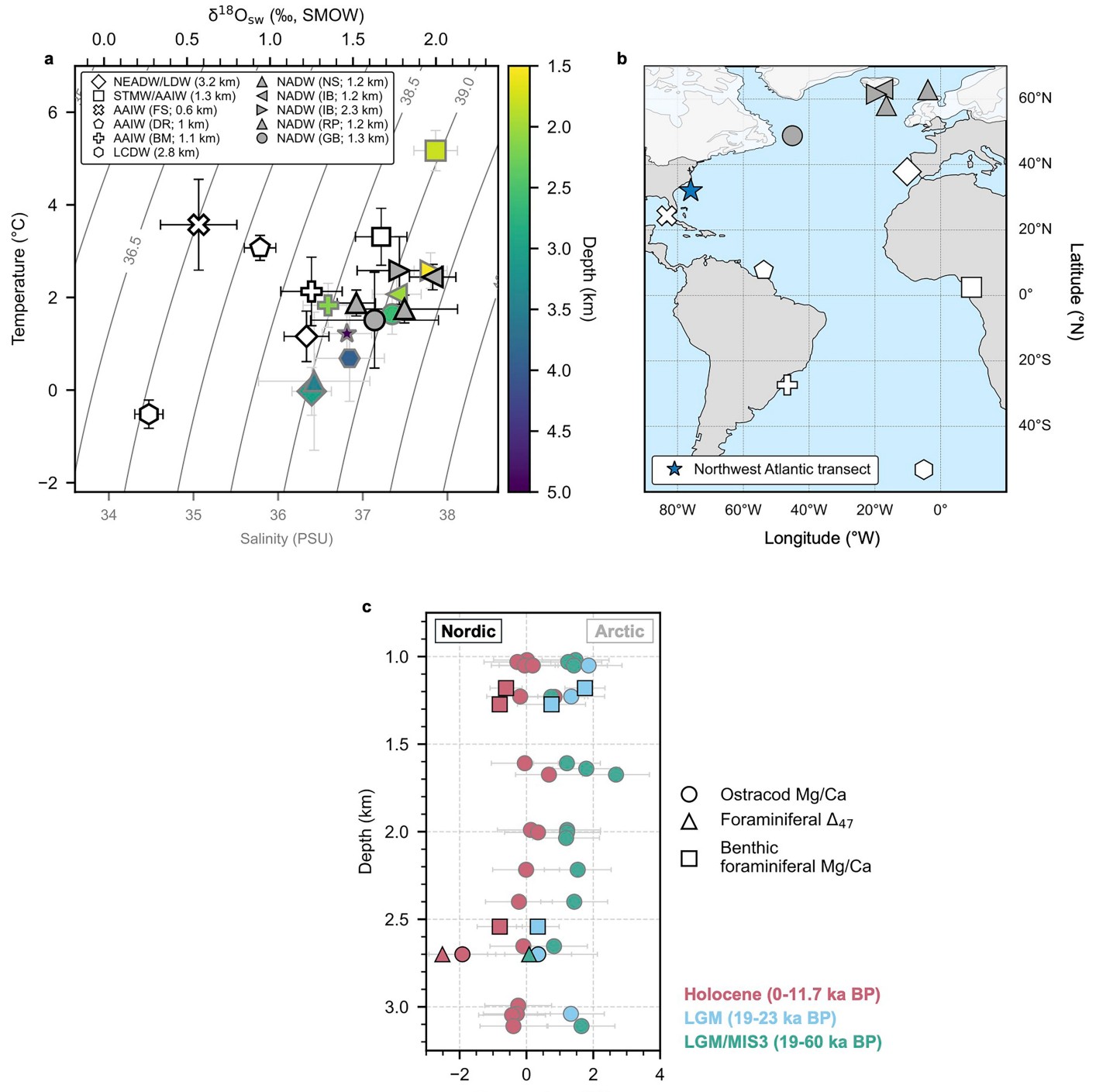

**Extended Data Fig. 5** | See next page for caption.

**Extended Data Fig. 5 | Comparison of Northwest Atlantic transect, Northeast Atlantic, and published Atlantic and Arctic Seas temperature and $\delta^{18}O_{sw}$ data. a**. Temperature-salinity cross plot as in Fig. 2f with additional published estimates of Atlantic glacial temperature and $\delta^{18}O_{sw}$, including associated ±1 s.e. (Methods). Data include cores located along the main pathway of modern NADW (open grey symbols; NS[77], IB, IB and RP (this study), GB[78]), as well as cores in locations influenced by other, non-NADW water masses (open white symbols; NEADW/LDW[79], STMW/AAIW[20], AAIW (FS)[74], AAIW (DR)[76], AAIW (BM)[75], and LCDW[80]). The former align closely with of our Northwest Atlantic data, consistent with our upstream tracing of NADW during the LGM. In comparison, the non-NADW cores are generally characterised by lower $\delta^{18}O_{sw}$. This suggests that (1) these sites were influenced by different glacial water masses, and (2) glacial NADW likely had higher $\delta^{18}O_{sw}$ than these other glacial water masses. One exception is Gulf of Guinea core MD03-2707 (open white square)[20], which exhibits relatively high $\delta^{18}O_{sw}$. This is expected, given the site's modern influence from a mix of AAIW and STMWs today, with the latter contributing to relatively high $\delta^{18}O_{sw}$. Notably, this core also lies on an approximate mixing line between AAIW and ODP-172-1055, which was bathed by STMWs during the LGM[14]. Iberian Margin core MD99-2234 (open white diamond)[79] and combined data from the Brazilian Margin (open white plus symbol)[75] also show relatively high $\delta^{18}O_{sw}$, likely reflecting the influence of Northeast Atlantic Deep Water in the Northeast Atlantic and southward flowing NADW and/or STMWs in the South Atlantic, respectively. **b**. Map showing the locations of cores corresponding to the published and Northwest Atlantic transect data presented in **a** (white shaded areas denote the approximate extent of the Laurentide Ice Sheet (LIS)[49] and Feno-Scandinavian Sheet (FIS) and British-Irish Sheet (BIS)[50] at 21.5 ka BP). **c**. Holocene and LGM deep ocean temperature reconstructions from the Nordic and Arctic Seas (black and grey edged symbols, respectively) derived from ostracod Mg/Ca[33] and benthic foraminiferal Mg/Ca[77,106,107] and benthic foraminiferal $\Delta_{47}$ (ref. 108; since data from the strict definition of the LGM is sparse, we also include ostracod data from last glacial period, i.e., Marine Isotope Stage 3 (MIS3)). Together, these records highlight that deep waters in the Nordic and Arctic Seas were warmer during the last glacial period than during the Holocene (Source Data). Associated error bars are ±2 s.e.

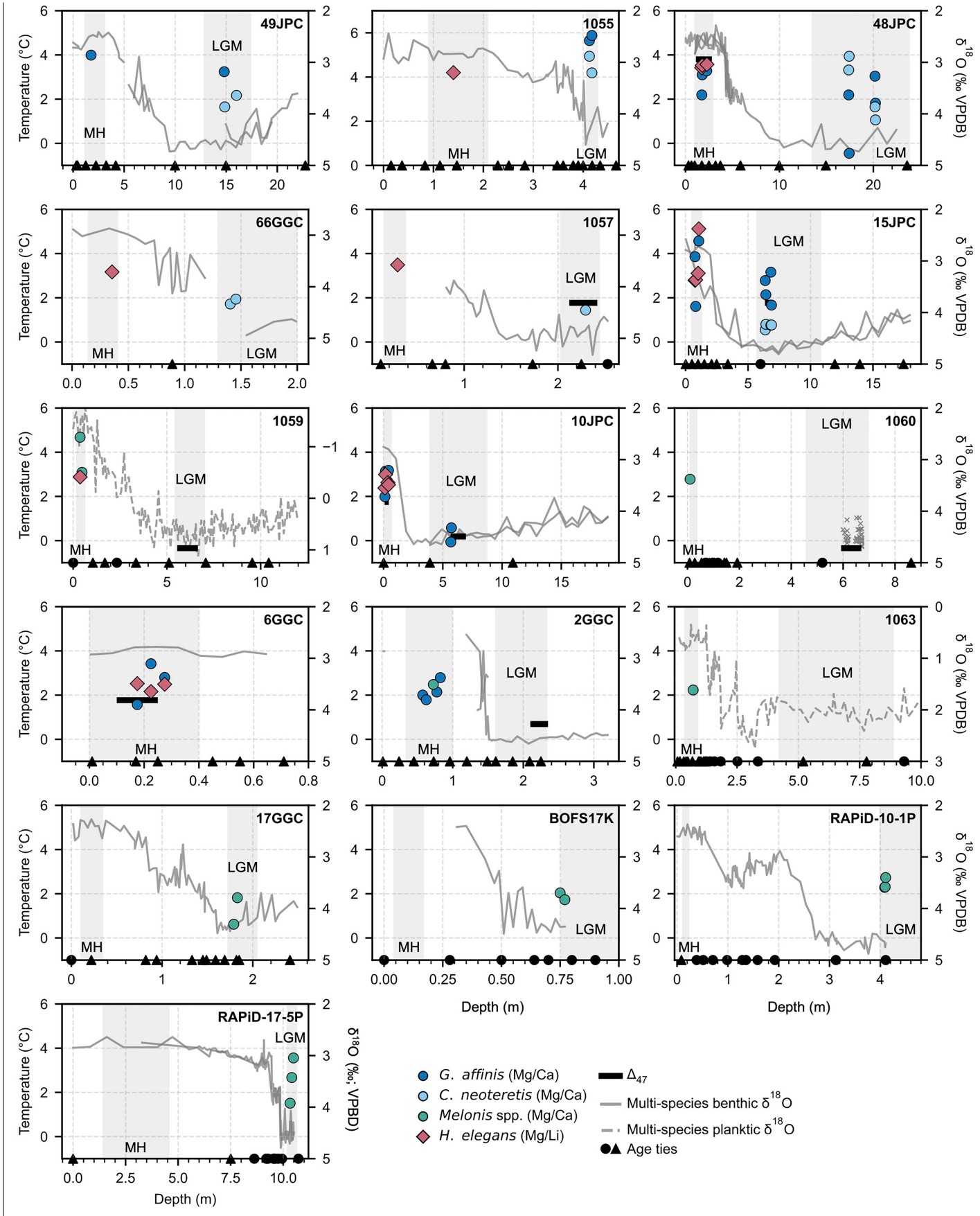

**Extended Data Fig. 6** | See next page for caption.

**Extended Data Fig. 6 | Stratigraphic overview of cores used in this study.** To highlight the relative downcore position of our new temperature constraints with respect to the mid-to-late Holocene and LGM, temperature proxy data are presented alongside downcore multi-species benthic or planktic foraminiferal $\delta^{18}O$ data, as well as each cores' corresponding age constraints. Most multi-species $\delta^{18}O$ data were obtained from published sources, including cores 49JPC, 48JPC, 15JPC, and 10JPC[14]; 1055, 1057, 1059, and 66GGC[109]; 17GGC[15]; RAPiD-10-1P and RAPiD-17-5P[51]. However, benthic and planktic foraminiferal $\delta^{18}O$ data from 6GGC, 2GGC, 1063, BOFS17K were generated as part of this study following the methods described in ref. 14 (Source Data). In the absence of downcore foraminiferal $\delta^{18}O$ data from core 1060, new glacial $\delta^{18}O$ (grey crosses) show that our $\Delta_{47}$ analyses were performed on samples corresponding to the glacial benthic $\delta^{18}O$ maxima. We also note that (1) all non-*Cibicidioides* spp. foraminiferal $\delta^{18}O$ data have been corrected for vital effects, apart from *U. peregrina* data from cores 1055, 66GGC, 1057, and 1060, and (2) both benthic and planktic foraminiferal $\delta^{18}O$ data were unavailable for cores 1055 and 1057; therefore, we use data from nearby cores KNR-140-2-51GGC and KNR-140-2-43GGC[109], which had been previously stratigraphically aligned[14]. Age constraints are taken from published age models (Extended Data Table 1) and comprise radiocarbon ages (filled black triangles) and stratigraphically aligned manual tie points (filled black circles). For each panel, grey shaded areas denote the approximate downcore position of the mid-to-late Holocene (left) and LGM (right). The thick black bars denote the depth range of samples used to generate $\Delta_{47}$ based temperature estimates, as well as the corresponding temperature estimate (for measurements where cores were combined, e.g., 1057/15JPC, the temperature shown for both cores represents the combined estimate).

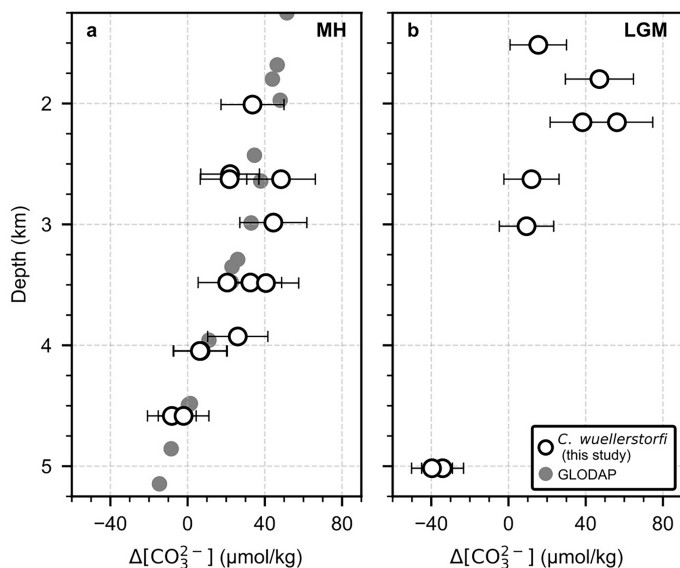

**Extended Data Fig. 7 | Reconstructed $\Delta CO_3^{2-}$ revealing the vertical $\Delta CO_3^{2-}$ structure of the deep Northwest Atlantic during the mid-to-late Holocene and LGM.** B/Ca derived $\Delta CO_3^{2-}$ for the mid-to-late Holocene (**a**) and LGM (**b**), and associated ±2 s.e. (2 s.e. = 2 × ($\Delta CO_3^{2-}$); methods). Each open circle denotes a single *C. wuellerstorfi* measurement, based on a sample comprising material from 5-15 specimens (open yellow circles). Modern Northwest Atlantic $\Delta CO_3^{2-}$ (grey circles) derived from modern hydrographic measurements from the GLODAP (v2.2022) database[47]. 0 μmol/kg denotes the threshold between oversaturated (>0 μmol/kg) and undersaturated (<0 μmol/kg) conditions[56].

**Extended Data Table 1 | Details of cores used in this study**

| Core | Lat. (N) | Long. (W) | Depth (km) | Time slice | MH temperature proxies | LGM temperature proxies | Age model |
|---|---|---|---|---|---|---|---|
| KNR-178-1-49JPC | 35.76 | 74.44 | 1.515 | MH/LGM | Mg/Ca | Mg/Ca | Ref. [14] |
| ODP-172-1055 | 32.78 | 76.28 | 1.798 | MH/LGM | Mg/Li | Mg/Ca | Ref. [14] |
| KNR-178-1-48JPC | 35.76 | 74.44 | 2.009 | MH/LGM | Mg/Ca; Mg/Li; $\Delta_{47}$ | Mg/Ca | Ref. [14] |
| KNR-140-2-66GGC | 32.50 | 76.29 | 2.155 | MH/LGM | Mg/Li | Mg/Ca | Ref. [109] |
| ODP-172-1057 | 32.03 | 76.05 | 2.584 | MH/LGM | Mg/Li | Mg/Ca; $\Delta_{47}$ | Ref. [14] |
| KNR-178-1-15JPC | 35.93 | 74.11 | 2.626 | MH/LHM | Mg/Ca; Mg/Li; $\Delta_{47}$ | Mg/Ca; $\Delta_{47}$ | Ref. [14] |
| ODP-172-1059 | 31.68 | 75.42 | 2.985 | MH/LGM | Mg/Ca; Mg/Li | $\Delta_{47}$ | Ref. [14] |
| KNR-178-1-10JPC | 35.84 | 73.89 | 3.015 | MH/LGM | Mg/Ca; Mg/Li; $\Delta_{47}$ | Mg/Ca; $\Delta_{47}$ | Ref. [14] |
| ODP-172-1060 | 30.77 | 74.47 | 3.481 | MH/LGM | Mg/Ca | $\Delta_{47}$ | Ref. [110] |
| KNR-178-1-6GGC | 35.82 | 73.59 | 3.485 | MH | Mg/Ca; Mg/Li; $\Delta_{47}$ | | Ref. [14] |
| KNR-178-1-2GGC | 36.12 | 72.29 | 3.927 | MH/LGM | Mg/Ca | $\Delta_{47}$ | Ref. [15] |
| ODP-172-1063 | 33.68 | 57.62 | 4.585 | MH | Mg/Ca | | Ref. [111,112] |
| KNR-197-10-17GGC | 36.42 | 48.54 | 5.011 | LGM | | Mg/Ca | Ref. [15] |
| BOFS-17K | 58.00 | 16.51 | 1.150 | LGM | | Mg/Ca | Ref. [52] |
| RAPiD-10-1P | 62.98 | 17.59 | 1.237 | LGM | | Mg/Ca | Ref. [51] |
| RAPiD-17-5P | 61.47 | 19.53 | 2.303 | LGM | | Mg/Ca | Ref. [51] |

The following references are cited in this table: refs. 14,15,51,52,109–112.

**Extended Data Table 2 | Species-specific common calibration intercepts, assuming a Mg/Ca-temperature sensitivity of 0.10 ± 0.003 mmol/mol/°C**

| Species | Intercept (±SE) |
|---|---|
| *Melonis* spp. | 0.82 ± 0.02 |
| *Bulimina* spp. | 1.03 ± 0.03 |
| *C. lobatulus* | 1.27 ± 0.03 |
| *N. labradorica* | 1.32 ± 0.03 |
| *C. neoteretis* | 0.82 ± 0.05 |

**Extended Data Table 3 | Details of cores and Mg/Ca-temperature calibrations used to derive estimates of glacial δ¹⁸O<sub>sw-ivc</sub> shown in Fig. 5**

Extended Data Table 3 | Details of cores and Mg/Ca-temperature calibrations used to derive estimates of glacial $\delta^{18}O_{sw-ivc}$ shown in Fig. 5

| Core | Location* | Lat. (N) | Long. (W) | Depth (km) | Foraminifera | Mg/Ca and $\delta^{18}O_c$ data reference | Mg/Ca-T calibration | Calibration reference |
|---|---|---|---|---|---|---|---|---|
| KNR-140-51GGC | Blake Outer Ridge (1) | 32.78 | 76.28 | 1.79 | *G. ruber* | Ref. [113] | Mg/Ca=exp(0.036 *(S-35)+0.064*T-0.87*(pH-8)-0.03) | Ref. [92] |
| KNR-140-39GGC | Blake Outer Ridge (1) | 31.67 | 75.32 | 2.98 | *G. ruber* | Ref. [114] | Mg/Ca=exp(0.036 *(S-35)+0.064*T-0.87*(pH-8)-0.03) | Ref. [92] |
| OCE-326-GGC5 | Bermuda Rise (2) | 33.70 | 57.58 | 4.55 | *G. inflata* | Ref. [113] | Mg/Ca=0.299*exp [0.009*(T-0.61*D)] [†] | Ref. [113] |
| MD01-2461 | Porcupine Seabight (3) | 51.75 | 12.92 | 1.15 | *G. bulloides* | Ref. [115] | Mg/Ca=exp(0.036 *(S-35)+0.064*T-0.88*(pH-8)+0.15) | Ref. [92] |
| ODP-162-980 | Feni Drift (4) | 55.48 | 14.65 | 2.18 | *N. pachyderma* | Ref. [116] | Mg/Ca=0.52* exp[T*0.10] | Ref. [117] |
| RAPiD-4P | South Iceland Rise (5) | 62.30 | 17.13 | 1.24 | *N. pachyderma* | Ref. [51] | Mg/Ca=0.52* exp[T*0.10] | Ref. [117] |
| JM11-F1-19JPC | Nordic Seas (6; surface) | 62.82 | 3.87 | 1.18 | *N. pachyderma* | Mg/Ca[77]; $\delta^{18}O_c$[118] | Mg/Ca=0.52* exp[T*0.10] | Ref. [117] |
| JM11-F1-19JPC | Nordic Seas (6; deep) | 62.82 | 3.87 | 1.18 | *M. barleeanum/C. neoteretis* | Ref. [77] | Common calibration | This study |
| RAPiD-10-1P | South Iceland Rise (7) | 62.98 | -17.59 | 1.24 | *M. barleeanum* | This study | Common calibration | This study |
| RAPiD-17-5P | South Iceland Rise (8) | 61.47 | -19.53 | 2.30 | *M. barleeanum* | This study | Common calibration | This study |
| BOFS17K | Rockall Plateau (9) | 58.00 | -16.51 | 1.15 | *M. barleeanum* | This study | Common calibration | This study |
| EW9302-2JPC | Grand Banks (10) | 48.80 | -45.09 | 1.25 | *M. barleeanum/C. lobtaulus* | Ref. [78] | Common calibration | This study |

*Numbers in parentheses correspond to the site number shown in Fig. 5. Site 1 is an average of both Blake Outer Ridge sites.
†D represents the core water depth in km and is included to account for potential dissolution effects. Data from refs. 51,77,78,92,113–118.