## [Peer Review File · Nature]

Relatively warm deep water formation persisted in the Last Glacial Maximum

Corresponding Author: Dr Jack Wharton

Version 0:

Reviewer comments:

Referee #1

(Remarks to the Author)

In this study, Wharton et al. present multiproxy-derived temperature and $d_{18}O_{sw}$ reconstructions across depth transects in the North Atlantic for the late–mid Holocene and the Last Glacial Maximum (LGM). Their results indicate that glacial NADW was not as cold as generally assumed. If correct, this finding has significant implications for our understanding of NADW formation and the Atlantic Meridional Overturning Circulation.

What sets this study apart is the thoughtful study design and the high quality of both the data and its presentation. I am particularly impressed by the comprehensive approach: depth transects across the North Atlantic, multiple geochemical proxies based on benthic foraminifera, and careful cross-comparison of these datasets. The reconstructions from different proxy types agree within error, and the authors' strategy of averaging them to derive mean temperature and $d_{18}O_{sw}$ increases robustness. Each proxy has its own limitations, but averaging them represents a constructive way forward. Importantly, the authors first validate the proxies by comparison with modern data, which further strengthens confidence in the approach.

Although this is not the first study to reconstruct deep-sea temperatures in the North Atlantic (the authors appropriately cite previous studies and use them for comparison, e.g., ostracode and pore-water data), the present work provides especially compelling evidence that challenges earlier reconstructions—particularly those based on pore-water chemistry (Adkins et al., 2002, *Science*). Overall, the conclusions are original, well supported by the data, and clearly articulated. The manuscript is exceptionally well written and accessible.

Given the central role of NADW in global ocean circulation, these results will be of wide interest to researchers in paleoclimatology, oceanography, and climate science. The data and presentation are both of very high quality. I do note, however, that some methodological sections could be clarified to improve transparency and reproducibility (see specific comments below). These issues do not undermine the validity of the conclusions. Proxy uncertainties are carefully addressed, and error bars are clearly defined.

For these reasons, I recommend publication pending minor revisions. This study will make a timely and valuable contribution to the community, and will serve as a benchmark for Earth system models used in future projections.

Lines 110–111: I agree with the approach, as the data from different species are comparable. For clarity, however, it would be useful to add a brief sentence explaining why it is reasonable to average multi-species data.

Line 270: “didn’t” please make sure that contractions are consistent with the journal’s style.

Line 298: Remove “clear”.

Line 333: If there is an upper size limit, please specify it here.

Lines 348–350: Provide a reference to the dataset or supplementary figures so readers can judge the reasoning for themselves.

Lines 387–400: Presumably this description refers only to trace element data? This is somewhat confusing because the preceding sentence mentions that both $\Delta 47$ and trace metal ratios were used in the study. Please clarify.

Line 403: Typo: “sensitives” → “sensitivities.”

Line 445: Typo: “the” → “then.”

Line 458: Report as “5.00 ‰.”

Lines 492–495: Some justification for this approach would be helpful. Is the same $d18O_{sw}$ –salinity relationship applicable to all sites? LeGrande and Schmidt (2006) is nearly 20 years old; consider citing or supporting the approach with more recent work.

Section “ ΔT and $\Delta d18O_{sw}$ ”: The anomalies were calculated using WOA data rather than Holocene proxy data to maximize coverage, since not all cores have both Holocene and LGM data. While understandable, it is generally preferable to calculate anomalies in proxy space to minimize proxy–climatology bias. To justify their approach, the authors could compare anomalies calculated using both WOA and Holocene data at sites where Holocene data exist.

Study location: Please provide a map showing the locations of the 14 sediment cores analyzed.

Referee #2

(Remarks to the Author)

Review of Wharton et al. submitted to Nature - 2025

The study “Thermal Structure of the Glacial Deep North Atlantic” investigates how the North Atlantic Ocean responded to glacial climate forcing during the Last Glacial Maximum (LGM). A key uncertainty in paleoclimate research is the thermal and salinity structure of the deep ocean, given the scarcity of direct constraints on past seawater properties. To address this, the authors reconstructed temperature and seawater $\delta^{18}O$ ($\delta^{18}O_{sw}$) using multi-proxy geochemical analyses from multiple sites in the North Atlantic Ocean. By constructing $\delta^{18}O_{sw}$ and temperature depth transects for the LGM North Atlantic, the study provides new insights into the persistence and characteristics of North Atlantic Deep Water (NADW) under glacial conditions. The results suggest that glacial NADW was only about 1.7–1.8 °C colder than today, significantly warmer than previously assumed, and carried a higher $\delta^{18}O_{sw}$ signature, linked to changes in the hydrological cycle and wind-driven gyre circulation.

Northwest Atlantic Temperature and $\delta^{18}O_{sw}$ Reconstructions

It is not obvious how the authors corrected $\delta^{18}O_{sw}$ for ice volume. A brief explanation would be useful here.

I appreciated that the authors did not over-interpret their results in terms of quantitative density. $\delta^{18}O_{sw}$ is strongly influenced by the hydrological cycle. I would strongly recommend checking whether the modern $\delta^{18}O_{sw}$ –salinity relationship is robust, or whether it reflects additional processes. Do we fully understand the role of the hydrological cycle in $\delta^{18}O_{sw}$ today? As the authors note, this region is complex, with deep-water formation potentially influenced by sea-ice formation and brine rejection, both of which affect $\delta^{18}O_{sw}$.

A comparison with simulated residual $\delta^{18}O_{sw}$ corrected for ice would be valuable (e.g., Caley et al., 2014). While neither models nor data are perfect, such a comparison would strengthen the interpretation, particularly regarding the structure of the water masses and, importantly, the relationship between $\delta^{18}O_{sw}$ and temperature.

Sustained Glacial Deep Water Production

As the authors acknowledge, Mg/Ca-based reconstructions can be biased, particularly when applied to planktonic foraminifera in regions where salinity is important. They use ice corrected $\delta^{18}O_{sw}$ derived from a wide range of sources, including planktonic foraminifera from subtropical to northeast sites along the Gulf Stream and North Atlantic Current. I assume that Mg/Ca is the primary thermometer used to estimate surface temperatures for $\delta^{18}O_{sw}$ reconstructions. The authors mention the use of updated calibrations but cite works that focus mainly on deep-water reconstructions. This raises several questions:

- Which calibration did they use for planktonic Mg/Ca?
- How did they correct for salinity effects? Local salinity varies substantially in this region, while existing corrections typically assume global open-ocean conditions (Gray et al., 2018).

Overall, I am not convinced by the surface-corrected $\delta^{18}O_{sw}$ reconstructions, which may be biased by circular reasoning: Mg/Ca estimates themselves are salinity-biased, yet are also used to correct $\delta^{18}O_{sw}$.

In addition, I assume that different species were used depending on site location. This complicates interpretation, as species typically found in northern regions often calcify deeper in the water column and may not reflect true surface conditions.

Clumped Isotope Method

I am unclear about the purpose of Extended Data Fig. 3a. Does it present the dataset of M21, or does it include additional data? If it is an updated calibration, essential methodological details such as the samples and estimated temperatures are missing. I strongly recommend calculating the regression parameters (slope, intercept, and associated uncertainties) using the methodology developed in Daëron and Vermeesch (2024). If it is M21, I do not see the point to show it.

I also do not understand the rationale for using the M21 calibration over others. For example, A21 and DV24 in Extended Data Fig. 3b yield values similar to M21 and GLODAP.

The sentence on line 953 is also unclear: why do the authors mention the need to correct for InterCarb only for P21?

Shell cleaning

How did the authors verify that the foraminiferal shells were clean? Did they check inside the shell for clay content? Did they take scanning electron microscope images for all species and levels at all sites studied? This is a critical point, as contamination can strongly affect $\delta^{18}\text{O}$ and $\delta^{13}\text{C}$ analyses.

Overall, the article is well written, but some parts are a bit difficult to follow. A few additional details would help readers better understand and properly follow the interpretations. For example, at line 203, a short example of assumptions now revised by pore water measurements would be valuable. In addition, the term Ocean is often missing after Atlantic.

In the data (rawD47), why don't some lines have D47 values? This seems strange to me. Is it due to a measurement issue?

Caley, T., Roche, D. M., Waelbroeck, C., and Michel, E.: Oxygen stable isotopes during the Last Glacial Maximum climate: perspectives from data–model (iLOVECLIM) comparison, *Clim. Past*, 10, 1939–1955, <https://doi.org/10.5194/cp-10-1939-2014>, 2014.

Gray, W. R., Weldeab, S., Lea, D. W., Rosenthal, Y., Gruber, N., Donner, B., & Fischer, G. (2018). The effects of temperature, salinity, and the carbonate system on Mg/Ca in *Globigerinoides ruber* (white): A global sediment trap calibration. *Earth and Planetary Science Letters*, 482, 607-620.

Referee #3

(Remarks to the Author)

The submission by Wharton et al. attempts to explore the bottom water temperatures in the North Atlantic and the Nordic Seas during the LGM using a variety of geochemical methods. In essence, they conclude that deepwater formation was a) continuous, and b) that bottom water temperatures were even warmer than today. While both findings are nothing really new as it has been suggested in various studies previously – and these already opposed the traditional narrative of strongly reduced deep convection – this new study now provides an excellent overview using sediment cores which cover a large region and a wide range of water depths. I realize that some of the authors have just very recently published on the topic in general and with similar conclusions (in Wharton et al. 2024; Blaser et al. 2025...etc). However, I regard this study as an excellent overview and summary of the state-of-the-art using paleoceanographic data. I therefore recommend publication but would like to urge the authors to consider the comments below.

- 1) According to the authors, their LGM is 4 kyrs long (23-19ka)... and this period often shows a trend in benthic ^{18}O - usually increasing towards 20ka -and certainly among the records from the different areas and depths worked with, too. So, I am missing the ages of the samples listed in the appended excel file, and in the paper please give a more precise age range for their(!) LGM.
- 2) Nordic Seas is the region of highest importance when it comes to deepwater formation processes today in general and for the LGM in particular. The cores used here, however, are actually from rather shallow southern sites and not necessarily suitable to study deeper bottom waters. Clearly it has been shown before that the deep Norwegian Sea provides LGM-data (MIS2) which already indicate enhanced bottom water temperatures (e.g., Thornalley et al. 2015; Cronin et al. 2012). It is somewhat a pity – considering the tremendous effort by the authors to compile a rather neat and comprehensive data set otherwise – not to have included also data from other, sites in the Nordic Seas where deepwater formation is actually occurring today. These are available from the deep Norwegian and even deeper Greenland Sea where bottom water T_s today range between -0.9 to -1.2°C and with a benthic shallow-infaunal species widely present during the LGM, i.e., *Oridorsalis umbonatus* (cf. tener). Using that one would be a real asset and provide a suitable end-member for the very coldest bottom water range. Besides, *O. umbonatus* is easily comparable with others, such *Cassidulina* (neo)teretis; btw, the latter genus is spelled with “ss”.
- 3) The ms covers many aspects and details which makes it hard to evaluate the data and to make out a stringent focus.
- 4) According to my impression the ms lacks more in-depth suggestions on what the likely process(es) was that lead to the relatively warm bottom waters. It has been clearly shown that subpolar planktonic foram species start to occur in the Nordic Seas with the onset of the LGM phase. Thus, I would to see the authors to also include a meaningful discussion of the

epibenthic $\delta^{13}\text{C}$ from that region. Although variable during the LGM, it would corroborate their own statement of a significant watermass contribution to the subpolar Atlantic from the northern region; see Blaser et. 2025 in which DJRT is a co-author too!

5) The way the paper is written and constructed there seems to be a better way in order to provide the reader with a somewhat easier access to the data set and its meaningful interpretations. As now, it is awkward to “juggle” between the text, figures, and supplementary data sheets.

6) With regard to points 3 and 6, I would urge the authors to draw another figure. This should include stratigraphic overview records of selected cores (vs. age or depth) showing the actual benthic ^{18}O records on the one hand and the measured temperature data on the other.

Version 1:

Reviewer comments:

Referee #1

(Remarks to the Author)

I previously reviewed an earlier version of this manuscript. I would like to thank the authors for their thoughtful revisions, which have further strengthened the paper. I find the authors' responses convincing and have no additional comments. I recommend the manuscript for acceptance. I would like to congratulate the authors on this solid piece of work, which will hopefully serve as a benchmark for future Earth system modeling studies.

Referee #2

(Remarks to the Author)

The authors have done an excellent job addressing my previous comments, and the revised version is now much clearer and more convincing. Their responses are thorough, and the manuscript has substantially improved. However, for this study to reach the level of robustness expected for publication, I believe one additional step is necessary, regarding shell cleaning. Although the authors explain that they relied on light microscopy and comparison with previously published isotope data to evaluate shell preservation, this is not fully sufficient to rule out contamination—especially given that biases affecting stable isotopes do not necessarily affect elemental ratios, and vice versa. I strongly encourage the authors to include a set of representative SEM images across species, depths, and sites to document shell preservation more rigorously if quantity of foraminifera allow it. Even a limited number of SEM checks (e.g., one or two specimens per species per site, few sites...) would greatly strengthen the confidence in their geochemical measurements and remove any remaining ambiguity regarding potential contamination.

With the addition of SEM documentation, I believe the manuscript will be fully satisfactory for publication.

Referee #3

(Remarks to the Author)

I have read the revised version of the manuscript by Wharton et al.. Overall my impression is that the authors have done a sufficient job to accommodate most of my comments, as well as those of the other 2 reviewers. With regard to the statistical background and validity of the employed deep-water temperature proxies there is no doubt that questions will remain to be asked, e.g., concerning their intra-comparability, foraminiferal calcite solubility issues, and to what extent the reconstructed “minor” temperature difference between the LGM and today, at the various sites studied, is credible and will hold during future testing. However, fundamental progress in understanding past conditions is only warranted if studies – such as this one – open up new gates for further thoughts and investigations. Therefore, I suggest the manuscript is good to go and should be published in Nature. Having said that, there are 2 minor things I have noticed which need to be looked at. In extended Data Fig. 6 the benthic $\delta^{18}\text{O}$ scale of site 1059 seems the wrong one – more likely $\delta^{13}\text{C}$, or is the record from planktic *G. ruber*!? In lines 229-232, the citation 41 listed describes brine formation as the sole cause, but the process of supercooling below ice shelves in such matters was actually coined in other palaeoceanographic publications (eg, Bauch et al 2001.). Please add accordingly.

Response to referees: “Thermal Structure of the Glacial Deep North Atlantic” by Jack Wharton, Emilia Kozikowska, Lloyd Keigwin, Thomas Marchitto, Mark Maslin, Martin Ziegler, and David Thornalley.

Author’s note – our detailed responses are given below in blue, with relevant direct quotes from the manuscript included where appropriate (“*blue italics*”). We have also included updated figures where appropriate.

Referee #1.

In this study, Wharton et al. present multiproxy-derived temperature and $\delta^{18}\text{O}_{\text{sw}}$ reconstructions across depth transects in the North Atlantic for the late–mid Holocene and the Last Glacial Maximum (LGM). Their results indicate that glacial NADW was not as cold as generally assumed. If correct, this finding has significant implications for our understanding of NADW formation and the Atlantic Meridional Overturning Circulation.

What sets this study apart is the thoughtful study design and the high quality of both the data and its presentation. I am particularly impressed by the comprehensive approach: depth transects across the North Atlantic, multiple geochemical proxies based on benthic foraminifera, and careful cross-comparison of these datasets. The reconstructions from different proxy types agree within error, and the authors’ strategy of averaging them to derive mean temperature and $\delta^{18}\text{O}_{\text{sw}}$ increases robustness. Each proxy has its own limitations, but averaging them represents a constructive way forward. Importantly, the authors first validate the proxies by comparison with modern data, which further strengthens confidence in the approach.

Although this is not the first study to reconstruct deep-sea temperatures in the North Atlantic (the authors appropriately cite previous studies and use them for comparison, e.g., ostracode and pore-water data), the present work provides especially compelling evidence that challenges earlier reconstructions—particularly those based on pore-water chemistry (Adkins et al., 2002, *Science*). Overall, the conclusions are original, well supported by the data, and clearly articulated. The manuscript is exceptionally well written and accessible.

Given the central role of NADW in global ocean circulation, these results will be of wide interest to researchers in paleoclimatology, oceanography, and climate science. The data and presentation are both of very high quality. I do note, however, that some methodological sections could be clarified to improve transparency and reproducibility (see specific comments below). These issues do not undermine the validity of the conclusions. Proxy uncertainties are carefully addressed, and error bars are clearly defined.

For these reasons, I recommend publication pending minor revisions. This study will make a timely and valuable contribution to the community, and will serve as a benchmark for Earth system models used in future projections.

We thank the reviewer for their supportive comments and have implemented their suggestions, as detailed below.

Lines 110–111: I agree with the approach, as the data from different species are comparable. For clarity, however, it would be useful to add a brief sentence explaining why it is reasonable to average multi-species data.

Additional text briefly justifying this approach has been added:

“The resultant temperature estimates from these different species are consistent and directly comparable, indicating no significant inter-species bias. We therefore averaged these multi-species data to derive mid-to-late Holocene and LGM mean temperature estimates for each core, reducing the overall uncertainty (Methods).”

Line 270: “didn’t” please make sure that contractions are consistent with the journal’s style. Amended.

Line 298: Remove “clear”. Removed

Line 333: If there is an upper size limit, please specify it here.

Due to the relative scarcity of benthic foraminifera in some samples, no upper size limit was applied during sieving. However, where possible, individuals of similar size were selected from each sample.

Lines 348–350: Provide a reference to the dataset or supplementary figures so readers can judge the reasoning for themselves. To aid transparency, the following text has been added:

“...these data were retained (individual measurements are provided in the Source Data).”

Lines 387–400: Presumably this description refers only to trace element data? This is somewhat confusing because the preceding sentence mentions that both Δ_{47} and trace metal ratios were used in the study. Please clarify.

We agree with the reviewer that this section could be made clearer. We have therefore reordered the Methods section and added clarifying text. Specifically, we moved the ‘Multi-species/multi-proxy approach’ paragraph to second in the Methods so that it precedes the trace metal based sections and subsequent \$\Delta_{47}\$ section. We then (1) updated lines 387 (referenced from the original manuscript and associated review) so that it is clear this text pertains to the averaging method used for the trace metal data, and (2) added additional text clarifying how cores from similar depths were combined for the \$\Delta_{47}\$ data (below).

“For sites represented by two cores from similar water depths (e.g., 1057/15JPC and 1059/10JPC), Δ_{47} values from both cores were averaged prior to temperature conversion.”

Line 403: Typo: “sensitives” → “sensitivities.” Corrected.

Line 445: Typo: “the” → “then.” Corrected.

Line 458: Report as “5.00 ‰.” Amended.

Lines 492–495: Some justification for this approach would be helpful. Is the same $\delta^{18}\text{O}_{\text{sw}}$ –salinity relationship applicable to all sites?

We agree with the reviewer that the use of a single \$\delta^{18}\text{O}_{\text{sw}}\$ –salinity relationship is a simplification, and we attempt to justify this choice in the main text and Methods. For example, in lines 130-132 and 150-154, we note that applying a uniform \$\delta^{18}\text{O}_{\text{sw}}\$ –salinity relationship results in a density inversion, implying that the North Atlantic was filled with multiple modes of NADW during the mid-to-late Holocene and LGM, formed at different sites and characterized by distinct \$\delta^{18}\text{O}_{\text{sw}}\$ –salinity relationships. The issue of selecting an appropriate \$\delta^{18}\text{O}_{\text{sw}}\$ –salinity relationship is particularly acute for the LGM when such relationships were likely different from those of the modern ocean. In the Methods (lines 498-504), we also emphasize our caution not to overinterpret the salinity estimates and instead focus primarily on the \$\delta^{18}\text{O}_{\text{sw}}\$ reconstructions. In the manuscript, this latter point was unclear, so we have revised the text as follows:

“While previous work has shown that different $\delta^{18}\text{O}_{\text{sw}}$ –salinity relationships apply at different depths in the Northwest Atlantic, likely reflecting distinct deep water masses sourced from different regions across the subpolar North Atlantic, for simplicity, we apply the NADW-specific relationship ($s = 0.51$, $c = 17.75$), as it is the most appropriate for the majority of our core sites, i.e., using this relationship yields the best agreement between our reconstructions and modern observations (Fig. 3c).”

LeGrande and Schmidt (2006) is nearly 20 years old; consider citing or supporting the approach with more recent work.

We appreciate the reviewer’s concern that the LeGrande & Schmidt (2006) \$\delta^{18}\text{O}_{\text{sw}}\$ –salinity relationship is now almost two decades old. Although recent studies have refined surface \$\delta^{18}\text{O}_{\text{sw}}\$ –salinity

relationships (e.g., Murray et al., 2023), new measurements aside (e.g., ‘LOCEAN’; Reverdin *et al.*, 2022), we are not aware of updates equivalent to the gridded product of LeGrande & Schmidt (2006) for the deep ocean. We therefore continue to use the relationships of LeGrande & Schmidt (2006) for our deep water reconstructions.

Section “ ΔT and $\Delta\delta^{18}O_{sw}$ ”: The anomalies were calculated using WOA data rather than Holocene proxy data to maximize coverage, since not all cores have both Holocene and LGM data. While understandable, it is generally preferable to calculate anomalies in proxy space to minimize proxy-climatology bias. To justify their approach, the authors could compare anomalies calculated using both WOA and Holocene data at sites where Holocene data exist.

We agree with the reviewer and thank them for this suggestion. We have now included the ΔT and $\Delta\delta^{18}O_{sw-ivc}$ values for both trace metal and Δ_{47} -based estimates (LGM minus mid-to-late Holocene) in the caption of Fig. 4. We have also updated the corresponding Methods text and added the paired core data to the Source Data file for Fig. 4.

From Fig. 5 caption: “We also calculated ΔT and $\Delta\delta^{18}O_{sw-ivc}$ for the LGM and mid-to-late Holocene using paired samples where available; the resulting estimates show close agreement with our broader climatological comparison: ($\Delta T_{Mg/Ca} = -1.6 \pm 0.7^\circ C$, $\Delta\delta^{18}O_{sw-ivc (Mg/Ca)} = 0.5 \pm 0.2 \text{ ‰}$, $n=5$; $\Delta T_{(\Delta 47)} = -2.2 \pm 2.0^\circ C$, $\Delta\delta^{18}O_{sw-ivc (\Delta 47)} = 0.4 \pm 0.5 \text{ ‰}$, $n=3$; Methods; Source Data).”

From Methods : “To minimize potential proxy-climatology bias, we also calculated ΔT and $\Delta\delta^{18}O_{sw}$ using paired LGM and mid-to-late Holocene data where available for both trace metal ($n=5$) and Δ_{47} ($n=3$) derived reconstructions. To do this, we followed the same procedure outlined above; however, for mid-to-late Holocene sites with both Mg/Ca- and Mg/Li-based temperatures, we used their combined mean values. For 6GGC, where only Holocene data are available, we compared these with LGM estimates from the nearby site 1060, which is also located at approximately 3.5 km water depth.”

Study location: Please provide a map showing the locations of the 14 sediment cores analyzed.

We agree with the reviewer that a map showing the locations of the cores used in this study would be useful, and we have therefore included this as Extended Data Fig. 1 (below).

“Extended Data Fig. 1. Map showing the location of cores used in this study. Northwest Atlantic (a) and Northeast Atlantic (b). Ocean bathymetry is from the GEBCO_2014 global bathymetric grid (30 arc-second resolution)⁹⁴.”

References:

Murray, N. K., Muñoz, A. R., & Conroy, J. L. (2023). Machine Learning Solutions to Regional Surface Ocean $\delta^{18}\text{O}$ -Salinity Relationships for Paleoclimatic Reconstruction. *Paleoceanography and Paleoclimatology*, 38(9), e2023PA004612.

Reverdin, G., Waelbroeck, C., Pierre, C., Akhoudas, C., Aloisi, G., Benetti, M., ... & Meredith, M. (2022). The CISE-LOCEAN seawater isotopic database (1998–2021). *Earth System Science Data*, 14(6), 2721-2735.

Referee #2:

Review of Wharton et al. submitted to Nature – 2025

The study “Thermal Structure of the Glacial Deep North Atlantic” investigates how the North Atlantic Ocean responded to glacial climate forcing during the Last Glacial Maximum (LGM). A key uncertainty in paleoclimate research is the thermal and salinity structure of the deep ocean, given the scarcity of direct constraints on past seawater properties. To address this, the authors reconstructed temperature and seawater $\delta^{18}\text{O}$ ($\delta^{18}\text{O}_{\text{w}}$) using multi-proxy geochemical analyses from multiple sites in the North Atlantic Ocean. By constructing $\delta^{18}\text{O}_{\text{sw}}$ and temperature depth transects for the LGM North Atlantic, the study provides new insights into the persistence and characteristics of North Atlantic Deep Water (NADW) under glacial conditions. The results suggest that glacial NADW was only about 1.7–1.8 °C colder than today, significantly warmer than previously assumed, and carried a higher $\delta^{18}\text{O}_{\text{sw}}$ signature, linked to changes in the hydrological cycle and wind-driven gyre circulation.

We thank the reviewer for their constructive comments, which we hope have helped to improve the clarity and readability of the manuscript.

Northwest Atlantic Temperature and $\delta^{18}\text{O}_{\text{sw}}$ Reconstructions

It is not obvious how the authors corrected $\delta^{18}\text{O}_{\text{sw}}$ for ice volume. A brief explanation would be useful here.

We agree with the reviewer that a description of how glacial $\delta^{18}\text{O}_{\text{sw-ivc}}$ was calculated would be useful; therefore, we have added the following text to the Methods:

“ $\delta^{18}\text{O}_{\text{sw-ivc}}$. To facilitate comparison with modern and mid-to-late Holocene $\delta^{18}\text{O}_{\text{sw}}$, we calculated ice volume corrected $\delta^{18}\text{O}_{\text{sw}}$ ($\delta^{18}\text{O}_{\text{sw-ivc}}$) for the LGM by subtracting 1.0 ‰ to account for the global ice volume effect. This correction is based on a global mean change in $\delta^{18}\text{O}_{\text{sw}}$ of 1.0 ± 0.1 ‰⁶⁹, derived from sedimentary pore water measurements, including non-Atlantic sites, which are considered more appropriate for estimating global mean glacial $\delta^{18}\text{O}_{\text{sw}}$ ¹⁷. Notably, this value also agrees well with other independent estimates (0.94 ± 0.18 ‰⁷⁰ and 1.05 ± 0.2 ‰⁷¹, based on simple numerical models of ice-sheet growth and benthic $\delta^{18}\text{O}$ change at polar sites, respectively). To further evaluate this correction, we also estimated the global mean $\delta^{18}\text{O}_{\text{sw}}$ change using the LR04 benthic foraminiferal stack⁷² and the most recent estimate of glacial MOT⁴⁶. Assuming a MOT change of 2.3 ± 0.5 °C and temperature sensitivity of 0.224 ‰/°C²⁷, a glacial-to-modern benthic foraminiferal $\delta^{18}\text{O}$ shift of 1.65 ‰, yields a global mean $\delta^{18}\text{O}_{\text{sw}}$ shift of 1.13 ± 0.11 ‰ (1 σ). This estimate is consistent, within uncertainty, with published values, supporting our use of the canonical 1.0 ‰ correction to derive $\delta^{18}\text{O}_{\text{sw-ivc}}$ for the LGM.”

I appreciated that the authors did not over-interpret their results in terms of quantitative density. $\delta^{18}\text{O}_{\text{sw}}$ is strongly influenced by the hydrological cycle. I would strongly recommend checking whether the modern $\delta^{18}\text{O}_{\text{sw}}$ –salinity relationship is robust, or whether it reflects additional processes. Do we fully understand the role of the hydrological cycle in $\delta^{18}\text{O}_{\text{sw}}$ today? As the authors note, this region is complex, with deep-water formation potentially influenced by sea-ice formation and brine rejection, both of which affect $\delta^{18}\text{O}_{\text{sw}}$.

We are pleased that the reviewer recognizes our caution in not over-interpreting our $\delta^{18}\text{O}_{\text{sw}}$ data in terms of salinity and density. We agree that modern $\delta^{18}\text{O}_{\text{sw}}$ –salinity relationships likely differed in the past due to differences in the hydrological cycle and glacial boundary conditions. Indeed, our dataset illustrates this complexity: glacial North Atlantic Bottom Water at ~5 km water depth (Fig. 3f) exhibits lower $\delta^{18}\text{O}_{\text{sw}}$ than the overlying water masses. For these reasons, rather than converting $\delta^{18}\text{O}_{\text{sw}}$ directly into salinity, we treat $\delta^{18}\text{O}_{\text{sw}}$ primarily as a conservative deep-ocean tracer, and use it to trace the pathway of relatively high $\delta^{18}\text{O}_{\text{sw}}$ from the Nordic Seas southwest into the deep Northwest Atlantic.

A comparison with simulated residual $\delta^{18}\text{O}_{\text{sw}}$ corrected for ice would be valuable (e.g., Caley et al., 2014). While neither models nor data are perfect, such a comparison would strengthen the interpretation, particularly regarding the structure of the water masses and, importantly, the relationship between $\delta^{18}\text{O}_{\text{sw}}$ and temperature.

We thank the reviewer for this suggestion. We agree that some consideration of previous modelling work strengthens our study, and we have added a paragraph comparing our $\delta^{18}\text{O}_{\text{sw}}$ reconstructions with the range of simulated $\delta^{18}\text{O}_{\text{sw}}$ values from six available glacial modelling studies. However, we remain cautious about over-interpreting these simulations, recognising potential model deficiencies, e.g., that the depth and structure of AMOC in many models are influenced by prescribed parameters and their resolution often hinders adequate simulation of e.g., the overflows (Hirschi et al., 2020).

From the main text: “We also compared our new $\delta^{18}\text{O}_{\text{sw-ivc}}$ reconstructions with the limited number of available isotope-enabled LGM simulations, which produce a relatively wide range of $\delta^{18}\text{O}_{\text{sw-ivc}}$ values for NADW (approximately -0.2 to 0.5 ‰; Methods and references therein). Of these, the iPOP2 simulation³² shows the best agreement with our proxy data, simulating high near-surface $\delta^{18}\text{O}_{\text{sw-ivc}}$ values in the western subtropical North Atlantic (2 ‰), which feed through into NADW at depth. However, neither STMW nor NADW extend as deep in the simulations as compared to proxy

reconstructions^{5,15}, likely due to limitations in the ability of models to simulate deepwater formation processes, in part linked to their spatial resolution³³.”

From Methods: “Isotope-enabled models. To provide additional context for our proxy reconstructions, we compared our new $\delta^{18}\text{O}_{\text{sw-ivc}}$ estimates with available isotope-enabled simulations of the glacial Atlantic^{32,84–88}. These models generally simulate a slightly shallower glacial AMOC, with NADW temperatures ranging from approximately 1 to -2°C ^{32,88} and $\delta^{18}\text{O}_{\text{sw}}$ between -0.2 and 0.5 ‰^{32,84–88}. While this is broadly consistent with the traditional view—based on paleoceanographic nutrient proxies³—that the AMOC shoaled during the LGM, and with pore-water-based estimates suggesting that glacial NADW was much colder⁶, these simulations are not consistent with more recent work⁵ and our new temperature and $\delta^{18}\text{O}_{\text{sw}}$ constraints from the North Atlantic. However, the iPOP2 model does reproduce comparably high $\delta^{18}\text{O}_{\text{sw}}$ values, although these are restricted to the upper ~ 2 km of the North Atlantic³².”

Sustained Glacial Deep Water Production

As the authors acknowledge, Mg/Ca-based reconstructions can be biased, particularly when applied to planktonic foraminifera in regions where salinity is important. They use ice corrected $\delta^{18}\text{O}_{\text{sw}}$ derived from a wide range of sources, including planktonic foraminifera from subtropical to northeast sites along the Gulf Stream and North Atlantic Current. I assume that Mg/Ca is the primary thermometer used to estimate surface temperatures for $\delta^{18}\text{O}_{\text{sw}}$ reconstructions. The authors mention the use of updated calibrations but cite works that focus mainly on deep-water reconstructions. This raises several questions:

- Which calibration did they use for planktonic Mg/Ca?

The planktic Mg/Ca-temperature calibrations used in this study were listed in the Source Data tab for Fig. 5 and Extended Data Fig. 4 (referenced to the original manuscript). However, we agree that this information was not sufficiently prominent. To improve transparency regarding how $\delta^{18}\text{O}_{\text{sw}}$ estimates were calculated, we have added an additional Extended Data Table (3) containing metadata and calibration data relevant to the $\delta^{18}\text{O}_{\text{sw}}$ reconstructions shown in Fig. 5. We have also updated the ‘Published Data’ section of the Methods to provide more detail on how planktic temperature and $\delta^{18}\text{O}_{\text{sw}}$ estimates were derived, and to clearly signpost Extended Data Table 3 and the relevant Source Data file, which also contains the published Mg/Ca and $\delta^{18}\text{O}_{\text{c}}$ glacial mean data.

“Glacial surface ocean temperature and $\delta^{18}\text{O}_{\text{sw}}$ estimates and associated uncertainties were also calculated from published Mg/Ca and $\delta^{18}\text{O}_{\text{c}}$ data using species-specific Mg/Ca temperature calibrations (Extended Data Table 3), appropriate vital-effect corrections^{89–91}, and the updated $\delta^{18}\text{O}$ -temperature relationship based on inorganic calcite precipitation experiments⁹¹ (Source Data). Although recent Mg/Ca-temperature calibrations for planktic foraminifera include non-thermal influences such as whole ocean salinity and pH, they do not account for spatial variations in local salinity⁹². For species with a known salinity effect on Mg/Ca (i.e. *G. ruber* and *G. bulloides*) we iteratively solved for a salinity value that is self-consistent with the North Atlantic $\delta^{18}\text{O}_{\text{sw}}$ -salinity relationship ($s = 0.55$, $c = 18.98$). For both *G. ruber* and *G. bulloides*, we assume a glacial pH of 8.2 ± 0.2 ^{92,93}, recognizing that this conservative uncertainty is the primary contributor to the relatively large errors on the final temperature and $\delta^{18}\text{O}_{\text{sw}}$ estimates (Source Data).”

- How did they correct for salinity effects? Local salinity varies substantially in this region, while existing corrections typically assume global open-ocean conditions (Gray et al., 2018). Overall, I am not convinced by the surface-corrected $\delta^{18}\text{O}_{\text{sw}}$ reconstructions, which may be biased by circular reasoning: Mg/Ca estimates themselves are salinity-biased, yet are also used to correct $\delta^{18}\text{O}_{\text{sw}}$.

We agree with the reviewer that local salinity conditions in the subtropical Northwest Atlantic likely varied during the LGM. As the reviewer notes, published Mg/Ca-temperature calibrations typically assume global open-ocean conditions. Therefore, in addition to calculating *G. ruber* temperatures using equation (1) from Gray and Evans, 2018, which includes a salinity sensitivity of 4 ‰ per PSU, we also accounted for potential local salinity variations by selecting an input salinity (the S term in Equation 1) that is self-consistent with the resulting value, using the North Atlantic $\delta^{18}\text{O}_{\text{sw}}$ -salinity relationship of

LeGrande and Schmidt, 2006. Importantly, the recalculated $\delta^{18}\text{O}_{\text{sw}}$ remains high ($\sim 0.9\text{‰}$), consistent with our other $\delta^{18}\text{O}_{\text{sw}}$ data from the glacial surface North Atlantic. Therefore, our conclusion that the elevated $\delta^{18}\text{O}_{\text{sw}}$ signal observed in the deep Northwest Atlantic was likely sourced from surface subtropical Northwest Atlantic via the Nordic Seas, remains robust. We have also updated Fig. 5, the corresponding Source Data, and the Methods text (see above) to describe this approach and its underlying assumptions.

In addition, I assume that different species were used depending on site location. This complicates interpretation, as species typically found in northern regions often calcify deeper in the water column and may not reflect true surface conditions.

We agree with the reviewer that species inhabiting deeper parts of the water column may not fully reflect true surface conditions, and therefore direct comparison of $\delta^{18}\text{O}_{\text{sw}}$ derived from species with different habitat depths could complicate interpretation. However, because the supply of NADW is fed by northward-flowing upper ocean waters spanning the top few hundred metres rather than strictly the surface layer (e.g., Burkholder and Lozier, 2024), the use of multiple planktic species to reconstruct $\delta^{18}\text{O}_{\text{sw}}$ along the Gulf Stream/North Atlantic Current pathway remains valid. For clarity, we have revised the text accordingly and also replaced the term ‘surface waters’ with either ‘near-surface’ or ‘upper ocean’ where appropriate.

“Fig. 5 shows how high $\delta^{18}\text{O}_{\text{sw-ivc}}$ —recorded consistently by multiple planktic foraminiferal species that occupy and reflect the properties of the subsurface upper ocean that is the source for NADW—is traceable from the western subtropical Atlantic, north-eastward along the path of the Gulf Stream and NAC into the likely deep water formation regions of the glacial subpolar North Atlantic and Nordic Seas, then back south at depth into the deep Northwest Atlantic.”

Clumped Isotope Method

I am unclear about the purpose of Extended Data Fig. 3a. Does it present the dataset of M21, or does it include additional data? If it is an updated calibration, essential methodological details such as the samples and estimated temperatures are missing. I strongly recommend calculating the regression parameters (slope, intercept, and associated uncertainties) using the methodology developed in Daëron and Vermeesch (2024). If it is M21, I do not see the point to show it.

Extended Data Fig. 3a (referenced in the original manuscript) originally presented the dataset from M21. We agree with the reviewer that this was an unnecessary addition and have removed it from the figure, replacing it with a comparison of our new oxygen isotope data and oxygen isotope data from the same suite of cores first presented in Wharton *et al.*, 2024, as part of our response to the reviewer’s later comment regarding shell cleaning.

“Extended Data Fig. 3. Comparison of $\delta^{18}\text{O}$ generated as part of this study with equivalent published data and Δ_{47} -based temperature estimates derived using different Δ_{47} -temperature calibrations. a. Kernel density estimate plots showing the distribution of $\delta^{18}\text{O}$ obtained from two independent analyses of benthic foraminifera from the same sediment samples for the mid-to-late Holocene and LGM (Δ_{47} , this study, blue; stable isotope analysis as per ref.¹⁵, teal). p -values from Welch’s t -tests indicate no significant difference between the two datasets (in both cases $p > 0.05$), confirming that the analyses yield indistinguishable $\delta^{18}\text{O}$. Close agreement between the published data and published $\delta^{18}\text{O}$ data from around the Northwest Atlantic suggests that the $\delta^{18}\text{O}$ measured in this study were unaffected by contamination, e.g., from clays. b and c. Comparison of mid-to-late temperature estimates derived using different published Δ_{47} -temperature calibrations: M21 (equation (1))⁶⁸, DG23 (equation (5))¹⁰⁰, A21 (equation (1))¹⁰¹, and DV24 (equation (28))¹⁰². Dashed lines in b indicate the approximate modern in-situ temperature at each core site, based on GLODAP 2022 data²⁹. The M21 calibration was selected for its strong statistical agreement with modern deep ocean temperatures and methodological consistency, despite being derived, in part, from planktic foraminiferal Δ_{47} data. This calibration was therefore applied to glacial samples as well.”

I also do not understand the rationale for using the M21 calibration over others. For example, A21 and DV24 in Extended Data Fig. 3b yield values similar to M21 and GLODAP.

We agree with the reviewer that our rationale for choosing the M21 calibration over others could be stated more clearly, and we thank them for highlighting this. To address this, we have added an additional panel (c) in the relevant Extended Data figure, showing the per-core difference from modern for each calibration, along with the mean and mean absolute errors. Both metrics demonstrate that M21 yields temperature estimates that provide the best statistical agreement with equivalent modern observational data. We have also updated the relevant Methods section and figure caption to highlight this statistical correspondence (Extended Data Fig. 3 is shown above).

The sentence on line 953 is also unclear: why do the authors mention the need to correct for InterCarb only for P21?

We thank the reviewer for identifying this inconsistency: P21 should indeed be P19. The study was published before the community-wide standardization of the clumped isotope paleothermometer (Bernasconi *et al.*, 2021), and thus, the mid-to-late Holocene would require correction before applying the P19 calibration. However, given that the P19-based measurements have been used in subsequent studies and that conversion to the updated reference frame is non-trivial, we have elected to omit this calibration from the manuscript to avoid introducing additional uncertainty.

Shell cleaning

How did the authors verify that the foraminiferal shells were clean? Did they check inside the shell for clay content? Did they take scanning electron microscope images for all species and levels at all sites studied? This is a critical point, as contamination can strongly affect $\delta^{18}\text{O}$ and $\delta^{13}\text{C}$ analyses.

We agree with the reviewer that contamination has the potential to affect $\delta^{18}\text{O}$ and $\delta^{13}\text{C}$ analyses. Therefore, prior to analysis, all samples were inspected under a light microscope to check for any visual contamination such as discoloured shells and the lack of typical shell surface textures. In addition, previous analyses on foraminifera from the same sediment samples have yielded excellent stable isotope and trace metal results, further indicating that the samples were not contaminated. To verify this independently, we compared our new $\delta^{18}\text{O}$ data with previously published $\delta^{18}\text{O}$ data measured on foraminifera from the same sediment samples. The two datasets show excellent agreement, as demonstrated by kernel density estimate plots and statistical tests (Welch's t-tests), which confirm no significant difference between them ($p > 0.05$; these plots and statistical results are included as a panel in Extended Data Fig. 3a and shown above). Furthermore, the published $\delta^{18}\text{O}$ data themselves are in very good agreement with other published data from around the Northwest Atlantic (Wharton *et al.*, 2024), providing confidence that both datasets accurately reflect in-situ conditions and were not affected by contamination (see figure below from Wharton *et al.*, 2024).

Fig. 2 | Vertical $\delta^{18}\text{O}$ profiles showing the structure of the ocean at Cape Hatteras and Blake Outer Ridge during the mid-to-late Holocene and LGM. **a**, Mid-to-late Holocene. **b**, LGM. Filled coloured circles and squares represent the mean value for each depth at Cape Hatteras and Blake Outer Ridge, respectively (individual monospecific batch data are shown in Extended Data Fig. 4). Associated error bars are ± 2 standard errors (standard error = σ/\sqrt{n} , in which σ is the multispecies replicate error at either Cape Hatteras or Blake Outer Ridge across both time periods and n is the number of species analysed at each depth). Open symbols correspond to published $\delta^{18}\text{O}$ from proximal sites in the

Northwest Atlantic $>35^\circ\text{N}$ (blue circles) and $<35^\circ\text{N}$ (red squares; Methods and Source Data)^{728,4748} and dashed coloured lines are smoothing splines through both new and published data $>35^\circ\text{N}$ (blue) and $<35^\circ\text{N}$ (red). All $\delta^{18}\text{O}$ are reported relative to the VPDB standard. The dashed horizontal black line in each panel denotes the change point at which both profiles first converge within error, which we interpret as the maximum depth of the STG. This was done through application of a piecewise linear regression model with one change point for all Blake Outer Ridge data from $>0.6\text{ km}$.

Overall, the article is well written, but some parts are a bit difficult to follow. A few additional details would help readers better understand and properly follow the interpretations. For example, at line 203, a short example of assumptions now revised by pore water measurements would be valuable.

We thank the reviewer for their supportive comments. We have added subheadings to improve readability and incorporated all reviewer suggestions, which we hope has enhanced the clarity and flow of the manuscript. Regarding the reviewer's specific comment, we agree that including an example of one of the assumptions underpinning the pore water method is important, and such an example has been included when introducing the proxy (lines 71-73 referenced to the original manuscript).

In addition, the term Ocean is often missing after Atlantic.

The omission of the term 'Ocean' is consistent with previous work recently published in Nature (e.g., Baker *et al.*, 2025; Wharton *et al.*, 2024). However, we are happy to use 'Atlantic Ocean' instead and defer to the preference of the Nature editorial team.

In the data (rawD47), why don't some lines have D47 values? This seems strange to me. Is it due to a measurement issue?

We thank the reviewer for their extremely thorough assessment of our manuscript. Yes, the absence of Δ_{47} data for some lines in the Source Data file reflects the exclusion of data and is part of the laboratory's standard screening protocol. This has been described in the Methods section: "The relationship between background signal and intensity was used to correct for non-linearities in isotopologues measurements. Samples with extreme initial intensities ($<11,000\text{ V}$ or $>20,000\text{ V}$) or high Δ_{47} standard deviations were excluded." These data were therefore intentionally omitted from the final dataset in accordance with our stated quality-control criteria, ensuring that only measurements within the linear range of the instrument are included, that standards have matching intensities, and that potential intensity-related biases in Δ_{47} are minimized. We also note that the omission of $\delta^{18}\text{O}$ measurements from samples lacking paired Δ_{47} data ($n = 143$) due to screening results in a mean $\delta^{18}\text{O}$ difference of $<0.01\text{‰}$; therefore, their exclusion does not bias the dataset.

Caley, T., Roche, D. M., Waelbroeck, C., and Michel, E.: Oxygen stable isotopes during the Last Glacial Maximum climate: perspectives from data–model (iLOVECLIM) comparison, *Clim. Past*, 10, 1939–1955, <https://doi.org/10.5194/cp-10-1939-2014>, 2014.

Gray, W. R., Weldeab, S., Lea, D. W., Rosenthal, Y., Gruber, N., Donner, B., & Fischer, G. (2018). The effects of temperature, salinity, and the carbonate system on Mg/Ca in *Globigerinoides ruber* (white): A global sediment trap calibration. *Earth and Planetary Science Letters*, 482, 607-620.

References:

Baker, J. A., Bell, M. J., Jackson, L. C., Vallis, G. K., Watson, A. J., & Wood, R. A. (2025). Continued Atlantic overturning circulation even under climate extremes. *Nature*, 638(8052), 987-994.

Bernasconi, S. M., Daëron, M., Bergmann, K. D., Bonifacie, M., Meckler, A. N., Affek, H. P., ... & Ziegler, M. (2021). InterCarb: A community effort to improve interlaboratory standardization of the carbonate clumped isotope thermometer using carbonate standards. *Geochemistry, Geophysics, Geosystems*, 22(5), e2020GC009588.

Burkholder, K. C., & Lozier, M. S. (2014). Tracing the pathways of the upper limb of the North Atlantic Meridional Overturning Circulation. *Geophysical Research Letters*, 41(12), 4254-4260.

Hirschi, J. J. M., Barnier, B., Böning, C., Biastoch, A., Blaker, A. T., Coward, A., ... & Xu, X. (2020). The Atlantic meridional overturning circulation in high-resolution models. *Journal of Geophysical Research: Oceans*, 125(4), e2019JC015522.

LeGrande, A. N., & Schmidt, G. A. (2006). Global gridded data set of the oxygen isotopic composition in seawater. *Geophysical research letters*, 33(12).

Wharton, J. H., Renoult, M., Gebbie, G., Keigwin, L. D., Marchitto, T. M., Maslin, M. A., ... & Thornalley, D. J. (2024). Deeper and stronger North Atlantic Gyre during the last glacial maximum. *Nature*, 632(8023), 95-100.

Referee #3:

The submission by Wharton et al. attempts to explore the bottom water temperatures in the North Atlantic and the Nordic Seas during the LGM using a variety of geochemical methods. In essence, they conclude that deepwater formation was a) continuous, and b) that bottom water temperatures were even warmer than today. While both findings are nothing really new as it has been suggested in various studies previously – and these already opposed the traditional narrative of strongly reduced deep convection – this new study now provides an excellent overview using sediment cores which cover a large region and a wide range of water depths. I realize that some of the authors have just very recently published on the topic in general and with similar conclusions (in Wharton et al. 2024; Blaser et al. 2025...etc). However, I regard this study as an excellent overview and summary of the state-of-the-art using paleoceanographic data. I therefore recommend publication but would like to urge the authors to consider the comments below.

We are grateful to the reviewer for their supportive comments and have responded to their suggestions, as detailed below.

1) According to the authors, their LGM is 4 kyrs long (23-19ka)... and this period often shows a trend in benthic $\delta^{18}\text{O}$ - usually increasing towards 20ka -and certainly among the records from the different areas and depths worked with, too. So, I am missing the ages of the samples listed in the appended excel file, and in the paper please give a more precise age range for their(!) LGM.

We agree with the reviewer that it is useful to illustrate the relative downcore position of our new temperature constraints with respect to downcore benthic foraminiferal $\delta^{18}\text{O}$ and/or other stratigraphic markers, particularly where the latter are unavailable. We have therefore added an additional Extended

Data figure showing, for each core, temperature versus depth for both the MH and LGM samples, together with previously published downcore benthic $\delta^{18}\text{O}$ data. The figure also includes the age constraints from each cores' age model, with the relative positions of the MH and LGM intervals marked (grey shading) accordingly.

As requested, we have also included ages for all individual (non-timeslice) temperature (Mg/Ca, Mg/Li, and Δ_{47}) and ΔCO_3^{2-} reconstructions in the Source Data files.

“Extended Data Fig. 6. Stratigraphic overview of cores used in this study. To highlight the relative downcore position of our new temperature constraints with respect to the mid-to-late Holocene and LGM, temperature proxy data are presented alongside downcore multi-species benthic or planktic foraminiferal $\delta^{18}\text{O}$ data, as well as each cores’ corresponding age constraints. Most multi-species $\delta^{18}\text{O}$ data were obtained from published sources, including cores 49JPC, 48JPC, 15JPC, and 10JPC¹⁵; 1055, 1057, 1059, and 66GGC¹⁰⁶; 17GGC¹⁶; RAPiD-10-1P and RAPiD-17-5P⁵¹. However, benthic and planktic foraminiferal $\delta^{18}\text{O}$ data from 6GGC, 2GGC, 1063, BOFS17K were generated as part of this study following the methods described in ref.¹⁵ (Source Data). In the absence of downcore foraminiferal $\delta^{18}\text{O}$ data from core 1060, new glacial $\delta^{18}\text{O}$ (grey crosses) show that our Δ_{47} analyses were performed on samples corresponding to the glacial benthic $\delta^{18}\text{O}$ maxima. We also note that (1) all non-Cibicidioides spp. foraminiferal $\delta^{18}\text{O}$ data have been corrected for vital effects, apart from *U. peregrina* data from cores 1055, 66GGC, 1057, and 1060, and (2) both benthic and planktic foraminiferal $\delta^{18}\text{O}$ data were unavailable for cores 1055 and 1057; therefore, we use data from nearby cores KNR-140-2-51GGC and KNR-140-2-43GGC¹⁰⁶, which had been previously stratigraphically aligned¹⁵. Age constraints are taken from published age models (Extended Data Table 1) and comprise radiocarbon ages (filled black triangles) and stratigraphically aligned manual tie points (filled black circles). For each panel, grey shaded areas denote the approximate downcore position of the mid-to-late Holocene (left) and LGM (right). The thick black bars denote the depth range of samples used to generate Δ_{47} -based temperature estimates, as well as the corresponding temperature estimate (for measurements where cores were combined, e.g., 1057/15JPC, the temperature shown for both cores represents the combined estimate).”

2) Nordic Seas is the region of highest importance when it comes to deepwater formation processes today in general and for the LGM in particular. The cores used here, however, are actually from rather shallow southern sites and not necessarily suitable to study deeper bottom waters. Clearly it has been shown before that the deep Norwegian Sea provides LGM-data (MIS2) which already indicate enhanced bottom water temperatures (e.g., Thornalley et al. 2015; Cronin et al. 2012). It is somewhat a pity – considering the tremendous effort by the authors to compile a rather neat and comprehensive data set otherwise – not to have included also data from other, sites in the Nordic Seas where deepwater formation is actually occurring today. These are available from the deep Norwegian and even deeper Greenland Sea where bottom water Ts today range between -0.9 to -1.2°C and with a benthic shallow-infaunal species widely present during the LGM, i.e., *Oridorsalis umbonatus* (cf. tener). Using that one would be a real asset and provide a suitable end-member for the very coldest bottom water range. Besides, *O. umbonatus* is easily comparable with others, such *Cassidulina* (neo)teretis; btw, the latter genus is spelled with “ss”.

We agree with the reviewer that Nordic Seas represent a key region for deep water formation, both today and during the LGM. Accordingly, we have expanded the section ‘Surface-deep ocean decoupling’ (starting on line 283 referenced to the new manuscript) to provide additional context on deep water formation processes in this region, where we explicitly describe the temperature changes previously reconstructed in the glacial Nordic and Arctic Seas, which we then link to our new Extended Data Fig. 5c. We also appreciate the reviewer’s recognition of our efforts to compile a comprehensive glacial deep North Atlantic temperature dataset, and we agree that our compilation would benefit from the inclusion of additional glacial data from the Nordic and Arctic Seas. To that end, we have incorporated published temperature estimates from multiple proxies—including *O. umbonatus* Mg/Ca ratios (Thornalley et al., 2015; Ezat et al., 2021), ostracod Mg/Ca (Cronin et al., 2012), and benthic foraminiferal clumped isotope measurements (Thornalley et al., 2015)—and now present these data as a depth transect as part of Extended Data Fig. 5c. Furthermore, to support our expanded discussion of deep water formation in the Nordic Seas, especially the Nordic Seas’ glacial warmth compared to the Holocene, we have also included equivalent Holocene data for each core where available.

“c. Holocene and LGM deep ocean temperature reconstructions from the Nordic and Arctic Seas (black and grey edged symbols, respectively) derived from ostracod Mg/Ca⁴⁰ and benthic foraminiferal Mg/Ca^{77,103,104} and benthic foraminiferal Δ_{47} (ref. ¹⁰⁵; since data from the strict definition of the LGM is sparse, we also include ostracod data from last glacial period, i.e., Marine Isotope Stage 3 (MIS3)). Together, these records highlight that deep waters in the Nordic and Arctic Seas were warmer during the last glacial period than during the Holocene (Source Data). Associated error bars are $\pm 2SE$.”

We have also corrected the spelling of *Cassidulina neoteretis*.

3) The ms covers many aspects and details which makes it hard to evaluate the data and to make out a stringent focus.

We have added subheadings, included additional Extended Data figures and tables, and reorganized sections to improve readability and focus, as well as incorporating all reviewer suggestions, which we think has helped to improve the manuscript’s clarity and flow, and helped to establish a more stringent focus.

4) According to my impression the ms lacks more in-depth suggestions on what the likely process(es) was that lead to the relatively warm bottom waters. It has been clearly shown that subpolar planktonic foram species start to occur in the Nordic Seas with the onset of the LGM phase. Thus, I would to see the authors to also include a meaningful discussion of the epibenthic $\delta^{13}C$ from that region. Although variable during the LGM, it would corroborate their own statement of a significant watermass contribution to the subpolar Atlantic from the northern region; see Blaser et. 2025 in which DJRT is a co-author too!

We agree with the reviewer that the manuscript could benefit from further discussion of the mechanisms that may have contributed to relatively warm glacial deep ocean temperatures and welcome the opportunity to provide some more discussion. Accordingly, we have expanded our text in the section ‘Surface-deep ocean decoupling’ to include additional discussion (starting on line 283 referenced to the new manuscript). With regard to the suggestion to include a meaningful discussion of epibenthic $\delta^{13}C$ from the Nordic Seas, we appreciate the reviewer’s point; however, we do not think a detailed treatment of regional $\delta^{13}C$ is appropriate within the scope of our study. Furthermore, the interpretation of $\delta^{13}C$ is influenced by multiple environmental and biogeochemical factors, and meaningful conclusions typically require a multi-proxy approach. Such an effort has already been undertaken by Blaser *et al.*, (2025), who compiled a large-scale, multi-proxy LGM dataset—including $\delta^{13}C$ —which they used to quantify the relative contributions of different deep-water masses to the North Atlantic. We have added additional text to highlight the results of this work and their consistency with our new temperature and $\delta^{18}O_{sw}$ data.

5) The way the paper is written and constructed there seems to be a better way in order to provide the reader with a somewhat easier access to the data set and its meaningful interpretations. As now, it is awkward to “juggle” between the text, figures, and supplementary data sheets.

In addition to introducing further subheadings and incorporating all reviewer suggestions to improve clarity and flow, we have also added some of the key source data associated with the sites and estimates shown in Figure 5 as an Extended Data Table (3), which includes the relevant calibration information to make the dataset and its interpretation easier to follow.

6) With regard to points 3 and 6, I would urge the authors to draw another figure. This should include stratigraphic overview records of selected cores (vs. age or depth) showing the actual benthic ^{18}O records on the one hand and the measured temperature data on the other.

As previously stated in response to comment (1), we agree with the reviewer’s comments, and we have included an additional Extended Data figure (ED fig. 6) showing downcore benthic foraminiferal $\delta^{18}\text{O}$ and temperature data.

References:

Cronin, T. M., Dwyer, G. S., Farmer, J., Bauch, H. A., Spielhagen, R. F., Jakobsson, M., ... & Stepanova, A. (2012). Deep Arctic Ocean warming during the last glacial cycle. *Nature Geoscience*, 5(9), 631-634.

Ezat, M. M., Rasmussen, T. L., Hain, M. P., Greaves, M., Rae, J. W., Zamelczyk, K., ... & Skinner, L. C. (2021). Deep ocean storage of heat and CO₂ in the Fram Strait, Arctic Ocean during the last glacial period. *Paleoceanography and Paleoclimatology*, 36(8), e2021PA004216.

Thornalley, D. J., Bauch, H. A., Gebbie, G., Guo, W., Ziegler, M., Bernasconi, S. M., ... & Yu, J. (2015). A warm and poorly ventilated deep Arctic Mediterranean during the last glacial period. *Science*, 349(6249), 706-710.

Response to referees: “Relatively warm deep water formation persisted in the Last Glacial Maximum” by Jack Wharton, Emilia Kozikowska, Lloyd Keigwin, Thomas Marchitto, Mark Maslin, Martin Ziegler, and David Thornalley. Author’s note – our responses are given below in blue.

Referee #1:

I previously reviewed an earlier version of this manuscript. I would like to thank the authors for their thoughtful revisions, which have further strengthened the paper. I find the authors’ responses convincing and have no additional comments. I recommend the manuscript for acceptance. I would like to congratulate the authors on this solid piece of work, which will hopefully serve as a benchmark for future Earth system modeling studies.

We thank the reviewer for their supportive response.

Referee #2.

The authors have done an excellent job addressing my previous comments, and the revised version is now much clearer and more convincing. Their responses are thorough, and the manuscript has substantially improved. However, for this study to reach the level of robustness expected for publication, I believe one additional step is necessary, regarding shell cleaning. Although the authors explain that they relied on light microscopy and comparison with previously published isotope data to evaluate shell preservation, this is not fully sufficient to rule out contamination—especially given that biases affecting stable isotopes do not necessarily affect elemental ratios, and vice versa. I strongly encourage the authors to include a set of representative SEM images across species, depths, and sites to document shell preservation more rigorously if quantity of foraminifera allow it. Even a limited number of SEM checks (e.g., one or two specimens per species per site, few sites...) would greatly strengthen the confidence in their geochemical measurements and remove any remaining ambiguity regarding potential contamination. With the addition of SEM documentation, I believe the manuscript will be fully satisfactory for publication.

We thank the reviewer for their supportive response, and as per their suggestion, to strengthen confidence in our geochemical measurements, we now include SEM photographs of representative benthic foraminifera as a Supplementary Information file. These SEM images show that the benthic foraminifera are well-preserved, with clearly defined ornamentation, whorls, sutures, open pores, and other surface features (Poirier *et al.*, 2021), and lack evidence of post-depositional alteration, i.e., authigenic carbonate precipitation an/or diagenetic overgrowths.

The SEM images are consistent with our two independent lines of geochemical evidence presented in the manuscript, i.e., (1) the agreement of our stable isotope data with published LGM records from coeval samples, and the consistency of these data with wider Northwest Atlantic \$\delta^{18}\text{O}\$ reconstructions (Wharton *et al.*, 2024; noting that previous work has shown any alteration affecting clumped isotope data typically also affects \$\delta^{18}\text{O}\$ (Leutert *et al.*, 2019)); and (2) for our Mg/Ca analyses, samples were rigorously cleaned following established cleaning protocols and then quantitatively screened for contaminant phases (as detailed in Methods), providing confidence in our Mg/Ca data.

References:

Leutert, T. J. et al., (2019). Sensitivity of clumped isotope temperatures in fossil benthic and planktic foraminifera to diagenetic alteration. *Geochimica et Cosmochimica Acta*, 257, 354-372.

Poirier, R. K. et al., (2021). Quantifying diagenesis, contributing factors, and resulting isotopic bias in benthic foraminifera using the foraminiferal preservation index: Implications for geochemical proxy records. *Paleoceanography and Paleoclimatology*, 36(5), e2020PA004110.

Wharton, J. H., et al., (2024). Deeper and stronger North Atlantic Gyre during the last glacial maximum. *Nature*, 632(8023), 95-100.

Referee #3.

I have read the revised version of the manuscript by Wharton et al.. Overall my impression is that the

authors have done a sufficient job to accommodate most of my comments, as well as those of the other 2 reviewers. With regard to the statistical background and validity of the employed deep-water temperature proxies there is no doubt that questions will remain to be asked, e.g., concerning their intra-comparability, foraminiferal calcite solubility issues, and to what extent the reconstructed “minor” temperature difference between the LGM and today, at the various sites studied, is credible and will hold during future testing. However, fundamental progress in understanding past conditions is only warranted if studies – such as this one – open up new gates for further thoughts and investigations. Therefore, I suggest the manuscript is good to go and should be published in Nature. Having said that, there are 2 minor things I have noticed which need to be looked at. In extended Data Fig. 6 the benthic d18O scale of site 1059 seems the wrong one – more likely d13C, or is the record from planktic *G. ruber*!?

We thank the reviewer for their supportive response. Yes, the new $\delta^{18}\text{O}$ record for core 1059 is from the planktic species *G. ruber*, as indicated in the figure legend, because downcore benthic $\delta^{18}\text{O}$ were unavailable for this core.

In lines 229-232, the citation 41 listed describes brine formation as the sole cause, but the process of supercooling below ice shelves in such matters was actually coined in other palaeoceanographic publications (eg, Bauch et al 2001.). Please add accordingly. Added.